# Improving AlphaFold2-based protein tertiary structure prediction with MULTICOM in CASP15

Jian Liu[1], Zhiye Guo [1], Tianqi Wu[1], Raj S. Roy[1], Chen Chen[1] & Jianlin Cheng [1✉]

Since the 14th Critical Assessment of Techniques for Protein Structure Prediction (CASP14), AlphaFold2 has become the standard method for protein tertiary structure prediction. One remaining challenge is to further improve its prediction. We developed a new version of the MULTICOM system to sample diverse multiple sequence alignments (MSAs) and structural templates to improve the input for AlphaFold2 to generate structural models. The models are then ranked by both the pairwise model similarity and AlphaFold2 self-reported model quality score. The top ranked models are refined by a novel structure alignment-based refinement method powered by Foldseek. Moreover, for a monomer target that is a subunit of a protein assembly (complex), MULTICOM integrates tertiary and quaternary structure predictions to account for tertiary structural changes induced by protein-protein interaction. The system participated in the tertiary structure prediction in 2022 CASP15 experiment. Our server predictor MULTICOM_refine ranked 3rd among 47 CASP15 server predictors and our human predictor MULTICOM ranked 7th among all 132 human and server predictors. The average GDT-TS score and TM-score of the first structural models that MULTICOM_refine predicted for 94 CASP15 domains are ~0.80 and ~0.92, 9.6% and 8.2% higher than ~0.73 and 0.85 of the standard AlphaFold2 predictor respectively.

[1] Department of Electrical Engineering and Computer Science, University of Missouri, Columbia, MO 65211, USA. ✉email: chengji@missouri.edu

Proteins carry out various functions such as catalyzing chemical reactions and regulating gene expression in living systems. The function of a single-chain protein is largely determined by its tertiary structure. Therefore, determining protein structures has been a major pursuit of the scientific community for decades. As low-throughput, expensive experimental techniques of determining protein structures such as x-ray crystallography can be only applied to determine the structures of a tiny portion of the proteins in nature. The computational prediction of protein structures from sequences holds the key of obtaining the structures for most proteins. With the decades of efforts of developing computational protein structure prediction methods, particularly the development of deep learning methods in the field[1–6], the accuracy of protein tertiary structure prediction has continued to improve and reached a high level in the last few years.

In the 2020 CASP14 experiment[7], AlphaFold2[8] predicted high-accuracy structures for most protein targets. Since then, it has become the most-widely used protein structure prediction tool. Despite its huge success, AlphaFold2 still cannot predict high-accuracy structures or even correct structural fold for some proteins that have very few or no homologous sequences in the existing protein sequence databases. Moreover, most existing protein structure prediction methods built on top of AlphaFold2 still do not take protein-protein interaction into account for protein tertiary structure prediction, even though the AlphaFold-Multimer work[4] has shown that considering the interaction between a protein and its partners in a protein complex is important to predict its tertiary structure.

In this work, we develop a new version of the MULTICOM protein structure prediction system to further improve AlphaFold2-based protein structure prediction by enhancing the input fed to AlphaFold2, using complementary approaches to rank AlphaFold2-generated structure models, and refining the top ranked models. Specifically, MULTICOM samples a set of diverse multiple sequence alignments (MSAs) from various sequence databases and identifies templates from different template databases as input for AlphaFold2 to generate more structural models, which increases the likelihood of AlphaFold2 generating high-quality structural models in the model sampling process. In addition to using the pLDDT[8] score that AlphaFold2 assign to each structural model to rank models, MULTICOM applies other complementary tertiary structure quality assessment methods such as APOLLO[9] of using the average similarity between a model and other models for the same target as quality score to rank them.

Moreover, MULTICOM introduces an iterative structure alignment-based refinement method to further improve the quality of the structural models. Specifically, it uses Foldseek[10] to search an input structural model against the protein structures in the Protein Data Bank (PDB)[11] and millions of predicted protein structures in the AlphaFoldDB[12] to find similar structures to augment the initial MSA used to generate the input model. The Foldseek-based structure alignment may find remote homologs or analogous proteins than the sequence alignment methods (e.g., HHblits) used by AlphaFold2 to enhance MSAs, which can be useful if the target has few homologous sequences. Moreover, Foldseek-based structure alignment can also find alternative structural templates that the sequence search may not find. The augmented MSA and alternative structural templates are used for AlphaFold2 to generate refined models.

Furthermore, for a monomer target that is a chain of a protein assembly, to consider its interaction with other protein chains in the assembly, MULTICOM uses our assembly/complex structure prediction method[13] built on top of AlphaFold-Multimer to predict the quaternary structure of the assembly first, and then it extracts the tertiary structure of the chain from the quaternary structure as a predicted structure of the monomer target.

MULTICOM participated in the tertiary structure prediction in the CASP15 experiment as both server and human predictors with different model ranking strategies, which ranked among the top CASP15 server and human predictors and performed much better than the standard AlphaFold2 predictor, demonstrating our approach of improving AlphaFold2-based protein tertiary structure prediction is effective.

## Results

**The comparison between MULTICOM servers and other CASP15 server predictors.** According to the CASP15 official ranking metric, MULTICOM_refine, MULTICOM_egnn, MULTICOM_deep and MULTICOM_qa ranked 3rd to 7th among server predictors (https://predictioncenter.org/casp15/doc/presentations/Day2/Assessment_TertiaryStructure_DRigden.pptx). The CASP15 official ranking metric for a model in a pool of models is the weighted average of the z-scores of multiple scoring metrics including lDDT[14], CADaa[15], SG[16], sidechain (side chain metrices like Average Absolute Accuracy (AAA)), MolProbity[17], backbone (backbone quality), DipDiff[18], GDT-HA[19], ASE (Accuracy Self Estimate based on the difference of pLDDT and lDDT), and reLLG[20] as shown in the formula $\frac{1}{16}(Z_{lDDT} + Z_{CADaa} + Z_{SG} + Z_{sidechain}) + \frac{1}{12}(Z_{MolPrb-clash} + Z_{backbone} + Z_{DipDiff}) + \frac{1}{6}(Z_{GDT-HA} + Z_{ASE} + Z_{reLLG})$. Such a score was calculated for no. 1 submitted model for each target by each predictor. The sum of the scores for all the CASP15 targets was the total score of a predictor, which was used to rank all the predictors. In addition to the no. 1 model, CASP15 also calculated the total score for the best of five models for the targets submitted by a predictor, which was used to rank the predictors alternatively.

To complement the CASP15 evaluation, in this analysis, we mainly use three widely scoring metrics—GDT-TS score, lDDT score, and TM-score—to evaluate MULTICOM predictors from different perspectives to elucidate their strengths and weaknesses.

The average GDT-TS scores of the top 20 CASP15 server predictors on 94 CASP15 domains are shown in Table 1. The standard AlphaFold2 predictor (NBIS-AF2-standard, AlphaFold v2.2.0) run by the Elofsson Group during the CASP15 experiment on all 94 domains ranks 20th. The 94 domains were extracted from 68 full-length tertiary structure prediction targets as individual assessment units by CASP15 organizers and assessors. Among the 94 domains, 47 domains are classified as Template-Based Modeling domains (TBM-easy or TBM-hard) whose structural templates can be identified among the known protein structures in the Protein Data Bank (PDB). The remaining 47 domains are classified as Free-Modeling (FM) domains that do not have a structural template or something between FM and TBM (i.e., FM/TBM) that may have some very weak template impossible for the existing sequence alignment methods to identify.

According to the average GDT-TS score on all 94 domains, MULTICOM server predictors (MULTICOM_refine, MULTICOM_egnn, MULTICOM_deep, MULTICOM_qa) ranked from 3rd to 6th among 47 server predictors and yielded largely similar performance. Our best performing server predictor - MULTICOM_refine ranks 3rd and has an average GDT-TS score of 0.7964, lower than 0.8433 of UM-TBM and 0.8397 of Yang-server but higher than all other predictors. Particularly, the average GDT-TS score of MULTICOM_refine is 9.3% higher than 0.7285 of NBIS-AF2-standard, indicating a notable improvement over the default AlphaFold2 has been achieved.

The average GDT-TS score of MULTICOM_refine on the 47 TBM domains is 0.8944, which is the same as that of Yang-server

and slightly lower than 0.9027 of UM-TBM, but higher than all the other servers (e.g., 8.3% higher than 0.8258 of NBIS-AF2-standard).

The main difference between MULTICOM servers and the top two servers (Yang-server and UM-TBM) lies in some harder FM and FM/TBM domains, particularly the ones involved in protein-protein interaction in protein assemblies. The average GDT-TS

**Table 1 The performance of the top 20 out of 47 server predictors in terms of the average GDT-TS of the top1 models (model 1) submitted by the server predictors for all 94 domains, 47 TBM domains and 47 FM and FM/TBM domains.**

| Predictor ID | Server name | Avg. GDT-TS on 94 domains | Avg. GDT-TS on 47 TBM domains | Avg. GDT-TS on 47 FM or FM/TBM domains |
|---|---|---|---|---|
| 62 | UM-TBM[40] | 0.8433 | 0.9027 | 0.784 |
| 229 | Yang-server[41] | 0.8397 | 0.8944 | 0.785 |
| 475 | MULTICOM_refine[42] | 0.7964 | 0.8944 | 0.6983 |
| 120 | MULTICOM_egnn[42] | 0.793 | 0.886 | 0.6999 |
| 158 | MULTICOM_deep[42] | 0.7922 | 0.8866 | 0.6977 |
| 86 | MULTICOM_qa[42] | 0.7917 | 0.8865 | 0.6969 |
| 35 | Manifold-E[43] | 0.7871 | 0.8821 | 0.692 |
| 166 | RaptorX[44] | 0.7808 | 0.8834 | 0.6782 |
| 288 | DFolding-server[45] | 0.7792 | 0.8755 | 0.6829 |
| 383 | server_124[46] | 0.7713 | 0.8866 | 0.6561 |
| 188 | GuijunLab-DeepDA[47] | 0.7646 | 0.8831 | 0.6462 |
| 298 | MUFold[48] | 0.7629 | 0.8783 | 0.6475 |
| 98 | GuijunLab-Assembly[49] | 0.7604 | 0.8746 | 0.6461 |
| 462 | MultiFOLD[50] | 0.7582 | 0.8767 | 0.6397 |
| 282 | GuijunLab-Threader[51] | 0.7578 | 0.869 | 0.6466 |
| 125 | UltraFold_Server | 0.7561 | 0.8782 | 0.6341 |
| 446 | Shennong | 0.7558 | 0.8732 | 0.6383 |
| 353 | hFold[52] | 0.7482 | 0.8635 | 0.633 |
| 466 | ColabFold[53] | 0.7352 | 0.796 | 0.6744 |
| 270 | NBIS-AF2-standard[54] | 0.7285 | 0.8258 | 0.6312 |

NBIS-AF2-standard is the standard AlphaFold2 predictor, which ranks 20th in terms of the average GDT-TS score on the 94 domains. MULTICOM_refine ranks 3rd.

score of the MULTICOM servers on the FM and FM/TBM domains is about ~0.7, which is lower than 0.7840 and 0.7850 of Yang-server and UM-TBM. Quite some difference comes from the 7 FM domains of six tertiary structure prediction targets (T1137s1-T1137s6) that are the subunits of one assembly target - H1137 (stoichiometry: A1B1C1D1E1F1G2H1I1) (see the native structure of H1137 in Supplementary Fig. 1a). The six chains A, B, C, D, E, and F (i.e., T1137s1 - T1137s6) of H1137 form a twisted helical transmembrane channel. The structure of the six chains needs to be predicted together in order to build a transmembrane channel to correctly predict the structure of the 7 FM domains in these chains. However, because the chains were released one by one for tertiary structure prediction during CASP15, our servers predicted the structures of the first several chains separately until after all the chains were released, resulting in the low-quality prediction for the FM domains in these chains. Because each of T1137s1 - T1137s6 is a long, non-globular helical structure that cannot be stabilized by itself, predicting the structure of one single chain without considering its interaction partners led to poor results. Moreover, AlphaFold-Multimer in our MULTICOM system predicted two kinds of transmembrane channels for the complex of the six chains: a largely straight one and a bended one (Supplementary Fig. 1b, c). The former one is much more similar to the native structure than the latter, whose FM domains have much higher GDT-TS scores than the latter. Unfortunately, when MULTICOM predicted the complex structure of all the chains after their sequences were released, the complex structure ranking method in MULTICOM selected the latter, leading to lower GDT-TS scores for the FM domains of some chains (e.g., T1137s6).

The performance of MULTICOM servers on the 68 full-length CASP15 targets in comparison with top CASP15 server predictors is reported in Supplementary Table 1. The average lDDT and TM-scores of MULTICOM servers and other top CASP15 predictors are shown in Supplementary Tables 2–5. The results in terms of lDDT and TM-scores are similar to those in terms of GDT-TS scores.

**The overall accuracy of MULTICOM_refine and its improvement over the standard AlphaFold2.** Figure 1a shows the distribution of TM-scores of the best of five models predicted by MULTICOM_refine for the 94 domains. MULTICOM_refine was able to predict the structure with the correct fold (TM-score > 0.5) for 85 domains (90.43%) out of 94 domains, 40 (85.11%) out

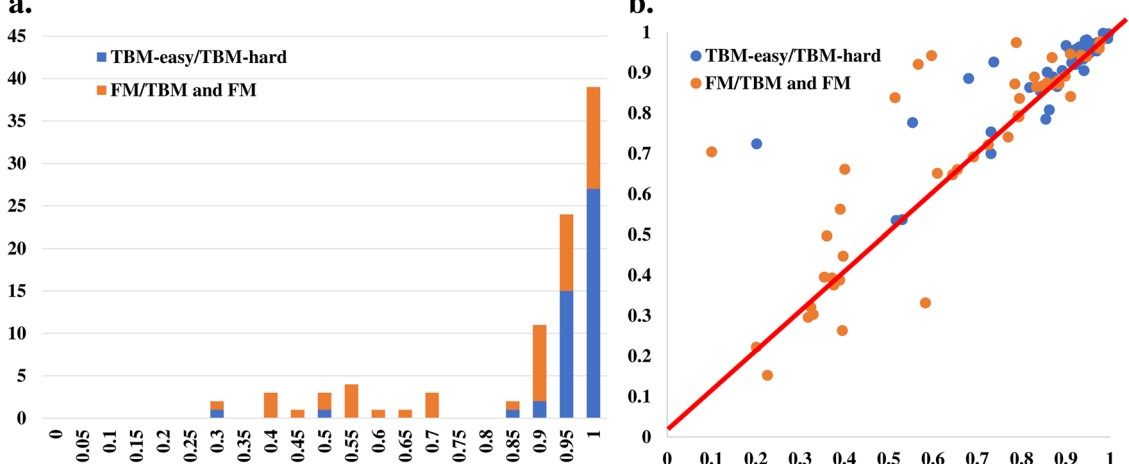

**Fig. 1 The overall qualities of the protein structures predicted by MULTICOM_refine. a** The histogram of the TM-scores of the best of five models predicted by MULTICOM_refine for 94 domains (47 FM and FM/TBM domains and 47 TBM domains). **b** The GDT-TS of the top 1 model predicted by MULTICOM_refine (y-axis) versus that of NBIS-AF2-standard (x-axis) on the 46 FM domains and 45 TBM domains.

**Table 2 The average GDT-TS scores of top 1 models or best of five models predicted for 91 domains by MULTICOM_refine and NBIS-AF2-standard.**

| Domain Type | Top 1 Model | | | Best of Five Models | | |
|---|---|---|---|---|---|---|
| | MULTICOM_refine | NBIS-AF2-standard | *p*-value | MULTICOM_refine | NBIS-AF2-standard | *p*-value |
| All 91 domains | 0.7922 | 0.7525 | 0.0006692 | 0.8185 | 0.7736 | 5.156e-06 |
| 46 FM & FM/TBM domains | 0.6939 | 0.6449 | 0.01347 | 0.7335 | 0.6746 | 0.0003845 |
| 45 TBM domains | 0.8927 | 0.8625 | 0.009202 | 0.9054 | 0.8749 | 0.001516 |

The *p*-value for the difference between MULTICOM_refine and NBIS-AF2-standard is calculated with the one-sided Wilcoxon signed rank test. In all the situations, MULTICOM_refine performs significantly better than NBIS-AF2-standard.

of 47 FM domains, and 45 (95.74%) out of 47 TBM domains. For 76 (80.85%) of 94 domains, 31 (65.96%) out of 47 FM domains, 45 (95.74%) out of 47 TBM domains, MULTICOM_refine predicted at least one high-accuracy model with TM-score > 0.8.

The only two TBM domains for which MULTICOM_refine failed to predict the correct fold are T1160-D1 (TM-score = 0.2825) and T1161-D1 (TM-score = 0.4546). Both of them are a domain of a single-domain protein chain of a homodimer, which has a small number of residues (29 and 48 residues, respectively), even though the CASP15 provided full-length 48-residue sequences of T1160 and T1161 that only differ in five residues. Despite the high sequence similarity, they fold into two different native conformations, making the structural prediction for the two domains hard. In fact, only one model for T1160-D1 or 11% of the models for T1161-D1 among all the models predicted by all the CASP15 predictors have TM-score > 0.5, even though the MSAs of the two targets contains hundreds of sequences, indicating that most predictors encountered difficulty to predict the structures for these two outlier targets using either AlphaFold2 or AlphaFold-Multimer.

For the FM and FM/TBM domains, MULTICOM_refine failed to predict a correct topology for T1122-D1, T1125-D5, T1125-D6, T1130-D1, T1131-D1, T1137s2-D2, T1137s3-D2, T1137s4-D2, T1137s6-D2 due to several different reasons. The reason that it failed on T1137s2-D2, T1137s3-D2, T1137s4-D2, T1137s6-D2 associated with the same complex (assembly) H1137 is mostly because the interaction between chains was not considered in the tertiary structure prediction or the incorrect quaternary structure for the complex was selected, which has been explained in Section 3.1. Predicting an accurate complex structure for H1137 is critical to obtain good structures for these domains. The similar problem happened to T1114s1, a subunit (chain A) of a large complex H1114 (stoichiometry: A4B8C8). In the native state, the four A chains interact tightly to form a cubic-like structure. However, the structure of T1114s1 was not predicted with all four A chains together by MULTICOM_refine, even though it was predicted with the presence of some B chains and C chains, leading to a low-quality model generated for it.

For T1122-D1, T1130-D1 and T1131-D1, very few or no homologous sequences and no significant structural templates could be found for them by the MSA sampling and monomer template identification, leading to lower TM-scores (TM-score=0.3705, 0.4372 and 0.2559, respectively) of the best models predicted for them. In this case, sampling many more models using AlphaFold2 with different parameters (e.g., a high number of recycles) may be able to generate better models.

T1125-D5 and T1125-D6 are two hard domains of a large 1200-residue monomer target T1125 consisting of six domains for which MULTICOM_refine did not generate good MSAs to cover the two domains, leading to poor prediction for them. Dividing this target into domains and predicting their structures

for the two domains separately may be able to generate better MSAs and tertiary structures for them.

Despite its failure on the domains above, MULTICOM_refine still performs significantly better than the standard AlphaFold2 predictor (NBIS-AF2-standard). Table 2 compares MULTICOM_refine and NBIS-AF2-standard in terms of average GDT-TS of the top 1 model on 91 common domains from 65 out of 68 full-length targets for which both NBIS-AF2-standard and MULTICOM_refine submitted predictions. Three domains (T1109-D1, T1110-D1 and T1113-D1) that NBIS-AF2-standard did not make predictions for were excluded from this analysis. The average GDT-TS of MULTICOM_refine of top 1 models predicted by MULTICOM_refine for the 91 domains is 0.7922, significantly better than 0.7525 of NBIS-AF2-standard with *p*-value = 0.0006692 according to one-sided Wilcoxon signed rank test. Similar results have been observed for the best of five models for the 91 domains, TBM domains, or FM and FM/TBM domains. The largely similar results on the 65 full-length targets are reported in Supplementary Table 6.

Figure 1b plots the per-domain GDT-TS score of the top 1 model predicted by MULTICOM_refine for each domain against that of NBIS-AF2-standard. MULTICOM_refine has a higher GDT-TS than NBIS-AF2-standard on 26 out of 46 FM domains (56.52%) and 28 out of 45 TBM domains (62.22%). On 12 domains, the top 1 model of MULTICOM_refine has a much higher GDT-TS than NBIS-AF2-standard (GDT-TS difference > 0.1), while on only two domains (T1122-D1 and T1125-D4) MULTICOM_refine has a much lower GDT-TS (GDT-TS difference < −0.1).

**The importance of sampling more models using AlphaFold2 with diverse MSA and template inputs**. The single-chain structure prediction shared by the MULTICOM servers used several different MSA and template inputs (Table 3) with AlphaFold2 to sample structural models. Therefore, it is important to evaluate if generating structural models with the diverse MSAs and templates input different from the default MSAs and templates used by AlphaFold2 can improve the accuracy of predicted structural models. All the MSA and template sampling methods in Table 3 except img and img_seq_temp only applied to a small number of targets are compared with NBIS-AF2-standard. The method of pooling the models from the different MSA and template sampling methods together is denoted as combine, which is the final method used by the single-chain structure prediction of the MULTICOM servers.

We compare the TM-scores of the best models that each sampling method generated for CASP15 full-length targets with those of NBIS-AF2-standard. During CASP15, the single-chain structure prediction shared by the MULTICOM servers was executed for 65 out of 68 full-length tertiary structure prediction

**Table 3 The 8 different kinds of MSAs, sequence databases, template databases and parameter settings for AlphaFold.**

| Name of MSA | MSA generation tool | Sequence database | Template database | AlphaFold2 network/ num_recycles/num_ensemble |
|---|---|---|---|---|
| default | HHblits, JackHMMER | UniRef30 and BFD, MGnify clusters | pdb70 | monomer/8/8 |
| default_seq_temp | HHblits, JackHMMER | UniRef30 and BFD, MGnify clusters | PDB_sort90 | monomer/8/8 |
| original | HHblits, JackHMMER | UniRef30, BFD, MGnify clusters | pdb70 | monomer/8/8 |
| ori_seq_temp | HHblits, JackHMMER | UniRef30, BFD, MGnify clusters | PDB_sort90 | monomer/8/8 |
| colabfold | MMseq2 | ColabFold DB | pdb70 | monomer/8/8 |
| colab_seq_temp | MMseq2 | ColabFold DB | PDB_sort90 | monomer/8/8 |
| img | DeepMSA | UniRef90, Integrated Microbial Genomes (IMG), metagenome sequence databases | pdb70 | monomer/8/8 |
| img_seq_temp | DeepMSA | UniRef90, Integrated Microbial Genomes (IMG), metagenome sequence databases | PDB_sort90 | monomer/8/8 |

Corresponding to each MSA, a set of structural templates is identified for a target from each of the two structure template databases (pdb70 or PDB_sort90) by using HHsearch to search the MSA against it. default and default_seq_temp (original and ori_seq_temp, colabfold and colab_seq_temp, or img and img_seq_temp) are the same MSA, but their corresponding templates are identified from the different template databases, leading to 8 different combinations of MSAs and templates. In all cases, the "monomer" network of AlphaFold2 is used to generate structural models. The num_recycles and num_ensemble are set to 8.

targets, except for the three targets (T1114s1, T1114s2, T1114s3) from a large complex H1114 due to the three-day time limit for server prediction. The three targets along with T1109, T1110 and T1113 which NBIS-AF2-standard did not make predictions for were excluded from this comparison. Moreover, if a sampling method only generated structural models for a subset of the targets, we compare it with NBIS-AF2-standard on the common targets which both of them made predictions for.

Figure 2a compares the TM-scores of the best models generated by each sampling method and NBIS-AF2-standard on the common set of targets. The average TM-score of the best models generated by the combine method for 62 common targets is 0.8105, significantly higher than 0.7838 of NBIS-AF2-standard ($p$-value = 1.182e−07) according to one-sided Wilcoxon signed rank test, demonstrating that using different MSAs and templates with AlphaFold2 to generate more models can significantly improve the quality of the best possible models.

The setting of the default sampling method in our single-chain structure prediction uses AlphaFold2's default MSA and template search programs to search the updated sequence databases (mgy_clusters_2022_05 and UniRef30_2021_03) and updated template databases (pdb70_from_mmcif_220313) to generate MSAs and templates for AlphaFold2 to predict 5 models with two higher parameter values (i.e., num_ensemble = 8 and num_recycles = 8). NBIS-AF2-standard also used the default preprocessing programs with the similarly updated databases to generate MSAs and templates for AlphaFold2 with the default parameter values to predict 5 models. The average TM-score of the best (or top1) models of our default sampling is 0.7913 (or 0.7766) on 62 common targets, only slightly higher than 0.7838 (or 0.77) of NBIS-AF2-standard. The difference is not significant as the $p$-value of one-sided Wilcoxon signed rank test is 0.1522 (or 0.4287), indicating that the slight change of the parameters of the default AlphaFold2 does not significantly change the quality of the best (or top1) models.

In contrast, the change of MSA and template inputs has a more significant impact on the quality of the best models generated for the CASP15 targets. Four out of six sampling methods that changed the input (i.e., default, default_seq_temp, original and ori_seq_temp) have higher average TM-score of best models than NBIS-AF2-standard. The $p$-value of the difference between three sampling methods (i.e., default_seq_temp, original and ori_seq_temp) and NBIS-AF2-standard is less than 0.05 (i.e., $p$-value = 0.04095, 0.01275, 0.0046, respectively), indicating a significant improvement was made. However, the two sampling methods (colabfold and colab_seq_temp) using the colab sequences databases (ColabFold DB) to generate MSAs have lower average TM-score of best models than NBIS-AF2-standard on the common targets, suggesting that their MSA quality may be somewhat lower than that of the MSAs of NBIS-AF2-standard on average. However, it is worth noting that the average TM-score of the best models generated by the four sampling methods together (i.e., default, default_seq_temp, original and ori_seq_temp) is 0.8065, lower than 0.8105 of pooling the models generated by all the individual MSA and template sampling methods in Table 3 in the single-chain structure prediction on the common targets, showing that although some sampling methods (i.e., colabfold and colab_seq_temp) cannot generate best models with higher TM-score than NBIS-AF2-standard on average, they may still generate some best models for some targets. The results show that increasing the diversity of MSAs and templates can improve the quality of the best models predicted by AlphaFold2.

The performance of the different sampling methods in terms of the top1 models ranked by AlphaFold2's pLDDT scores is shown in Fig. 2b. The average TM-score of the top1 models generated by the combine method for 62 common targets is 0.786, higher than

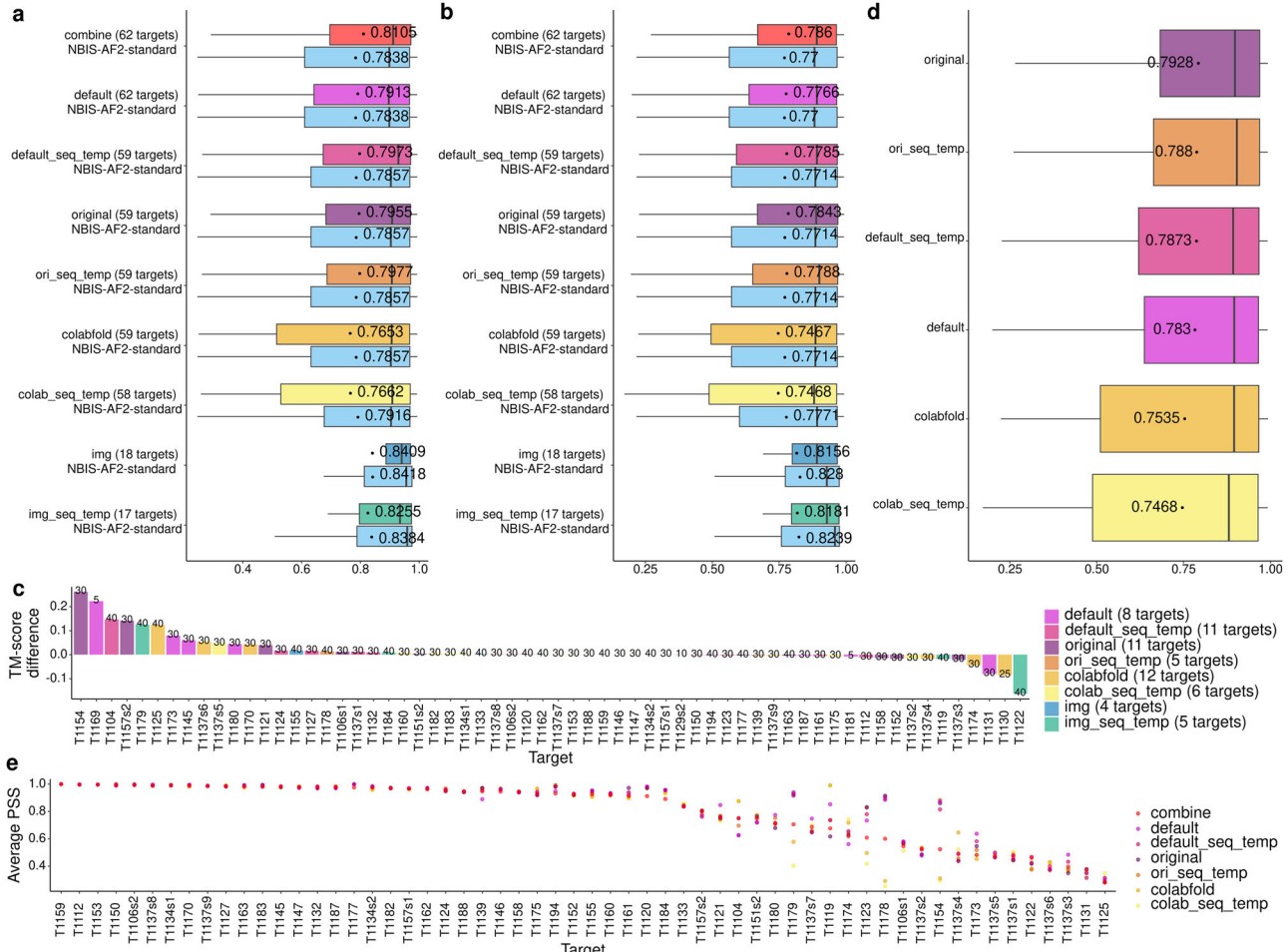

**Fig. 2 In-depth analysis of the sampling methods. a** The comparison between the TM-scores of the best models generated by each MSA and template sampling method and the NBIS-AF2-standard on the common full-length targets. The per-target mean (or median) TM-score of each sampling method and the NBIS-AF2-standard is located and reported by the black dot in the box (or located by a vertical line). **b** The comparison between the TM-scores of the top1 (rank1) models generated by each MSA and template sampling method and the NBIS-AF2-standard on the common full-length targets. The per-target mean (or median) TM-score of each sampling method and the NBIS-AF2-standard is located and reported by the black dot in the box (or located by a vertical line). **c** The TM-score difference between the top1 model selected by AlphaFold2 pLDDT score in combine are plotted against the top1 model of the NBIS-AF2-standard. The number of models in the combine are specified near the boxes for each target. **d** The comparison between the TM-scores of the top1 models generated by original, ori_seq_temp, default_seq_temp, default, colabfold and colab_seq_temp on the common 58 full-length targets ordered by the average top1 TM-scores. The per-target mean (or median) TM-score of each sampling method is located and reported by the black dot in the box (or located by a vertical line). **e** The average pairwise similarity score (PSS) of models from combine, default, default_seq_temp, original, ori_seq_temp, colabfold and colab_seq_temp on each of the 58 common targets. If the average PSS values are almost the same, the dots denoting the values for a target overlap. In this case, the average PSS of the models from combine (red dot) is plotted on the top covering the other dots denoting the almost same values.

0.77 of NBIS-AF2-standard without significant difference (p-value = 0.05938) according to one-sided Wilcoxon signed rank test. In Fig. 2c, the TM-score difference between the top1 model of the combine method selected by AlphaFold2's pLDDT score and the top1 model of NBIS-AF2-standard as well as the number of models of the combine method is reported for each target. On most targets, the difference is positive, i.e., the top1 model of the combine method has a higher TM-score than NBIS-AF2-standard. As shown in Fig. 2b, four sampling methods (i.e., default, default_seq_temp, original and ori_seq_temp) have higher average TM-score of top1 models than NBIS-AF2-standard on the common targets, but there is no significant difference between each of them and NBIS-AF2-standard.

To better compare the performance between the in-house sampling methods that generated models for most of the CASP15 targets, the average TM-scores of the top1 models from the six sampling methods are shown in Fig. 2d, ordered by their average

TM-score of the top1 models. The results of the one-sided Wilcoxon test indicate that there is no significant difference between original and ori_seq_temp, original and default_seq_temp in terms of the TM-score of the top1 models. However, there is significant difference between original and each of the other three sampling methods (default, colabfold, colab_seq_temp), with the p-value of 0.04604, 0.01678 and 0.01846 respectively.

To quantify the distribution of the similarity of the models generated by different methods, the average pairwise similarity score (PSS) of the models produced by each method is calculated. A higher average PSS indicates that the models in the model pool of a method are more similar, while a lower PSS suggests the presence of multiple or more diverse conformations in the model pool. To visualize the results, the average PSS of the models from combine, default, default_seq_temp, original, ori_seq_temp, colabfold and colab_seq_temp are plotted for each of the 58

common targets in Fig. 2e. In case the average values are almost the same, the dot denoting the average PSS of the combine model pool is always plotted at the top, which may cover the dots denoting the almost same values of the other methods. The figure shows that, for 33 out of 58 targets, the average PSS of the methods is greater than 0.9, indicating that they all generated models of similar/same conformations. However, for 22 out of 58 targets, the average PSS of the combine model pool is less than 0.8, suggesting there is a diverse set of models in the model pool. Interestingly, for T1104, T1179, T1119, T1123, T1178 and T1154, some methods generated very similar conformations (e.g., the average PSS of colabfold models generated for T1104 is 0.8751), while the other methods (e.g., default_seq_temp) generated models with more different conformations (the average PSS of default_seq_temp for T1104 is 0.6247). The diversity of the MSAs and templates used by the different methods increases the variety of models in the combine model pool for these targets. The average TM-score of top1 models of the combine method on the 33 targets that has PSS value greater than 0.9 is 0.8858, only slightly higher than 0.8823 of NBIS-AF-2-standard, indicating that the targets are mostly easy and generating more models from diverse MSAs and templates only have a small effect on the performance. However, on the 22 targets for which the PSS value of top1 models of the combine method is less than 0.8, the average TM-score of the top1 models of the combine is 0.6344, notably higher than 0.6073 of NBIS-AF2-standard, indicating that these targets are mostly harder ones and generating more models from diverse MSAs and templates has a larger effect on the performance. Indeed, for 14 out of the 22 targets, the top1 model in the combine model pool has a higher TM-score than that of NBIS-AF2-standard, showing that the increased variety of models improves the quality of the models for 63.64% of these targets.

**The effect of Foldseek structure alignment-based refinement method.** MULTICOM_refine used a Foldseek structure alignment-based iterative refinement method with AlphaFold2 to refine the top 5 models selected for a single-chain monomer target. In CASP15, the refinement method was only applied to 23 out of 27 single-chain monomer targets. It was not applied to the tertiary structural models extracted from predicted complex structures because refining tertiary structures alone without considering the interaction with the partners would likely make the structural models worse. Although the average GDT-TS of the refined models and the original models is not significantly different (0.7887 versus 0.7910) as shown in Supplementary Table 7, there are some cases where the refinement method significantly improved the quality of the initial model.

Figure 3 compares the initial model of T1180 (GDT-TS = 0.7322) and the refined model submitted by MULTICOM_refine (GDT-TS = 0.8951), which is the best model among all CASP15 models for this target predicted by all the CASP15 predictors. In this case, Foldseek was able to find four similar structures as templates, while the sequence-based template searching found only one of the four templates. Foldseek also found several structure alignments added into the MSA.

To investigate the factors that caused the improvement on T1180, we performed the following post-CASP15 experiments with AlphaFold2 to generate 15 models respectively, which was the total number of models produced/used during the refinement process. The refinement process used the five models of the "*default*" sampling method as initial models to generate 5 models in each of the three refinement iterations, resulting in 15 refined models in total for selection.

In the first experiment, the default sampling method was used to generate 15 models for T1180. The GDT-TS of the top1 model

is 0.8583, higher than 0.7322 of the top1 model among the five models initially generated by the default sampling method, but lower than 0.8951 of the final refined model from the refinement. The experiment indicates that increased sampling can yield models of higher quality, but still cannot reach the quality of the refinement process.

In the second experiment, the combined alignments in each iteration of the refinement process along with the sequence search-found templates were fed to AlphaFold2 to generate 15 models. The GDT-TS of the top1 models for each iteration is 0.666, 0.8661 and 0.8806. The final score of 0.8806 is very close to 0.8951 of the final refined model submitted to CASP15, demonstrating that iteratively adding Foldseek-found structure alignments into the MSA in the refinement process is a reason for the improvement in model quality.

In the third experiment, the four templates (3JS3A, 3JS3B, 3NNTB, 4H3DB) identified by Foldseek along with the initial MSA were provided to AlphaFold2 to generate 15 models. The GDT-TS of the top1 model of the refinement process is 0.71, much lower than 0.8951 of the final refined model submitted to CASP15, indicating that the structural templates is not the reason leading to the model quality improvement. Furthermore, we used one template 4H3DB four times as structural templates for AlphaFold2 to generate models, resulting in a top1 model with GDT-TS of 0.7231, much lower than 0.8951 of the final refined model submitted to CASP15. This further confirms that adding more templates into the AlphaFold2 model generation is not the reason that the refinement process produced the high-quality models in CASP15.

**The importance of considering protein-protein interaction in predicting the tertiary structure of a monomer that is a subunit of a protein assembly.** Out of 68 full-length tertiary structure prediction targets, 41 targets are a subunit of a protein assembly. The single-chain structure prediction for such a target cannot consider the interaction between the target and its interaction partners in the assembly, while the assembly structure prediction takes into account the interaction and is expected to generate better structural models for it.

Figure 4a compares the GDT-TS of the best model generated from the single-chain structure prediction with those of the best model extracted from the assembly structures generated by the assembly structure prediction on the 38 out of 41 targets, excluding the three subunits of H1114 (e.g., T1114s1, T1114s2, T1114s3) because full-length models for H1114 were not generated by the MULTIOM system during CASP15. The latter has higher GDT-TS than the former for 36 out of 38 targets (94.7%), validating that considering the protein-protein interaction (i.e., the folding context) is important to predict the tertiary structure of a target that interacts with other proteins to form an assembly.

Figure 4b, c shows how the assembly structure prediction considering protein-protein interaction generated a much better model than the single-chain structure prediction without considering protein-protein interaction for T1157s2 (a chain of heteromultimer H1157) and T1173 (a chain of homomultimer T1173o). For T1157s2, the GDT-TS of the tertiary structure model extracted from the assembly model is 0.7408, much higher than 0.5858 of the monomer model predicted by the single-chain structure prediction. Compared with the native structure (Fig. 4b), the main difference between the two models is in the leftmost (C-terminal) domain that interacts with other chains in H1157. Without considering the protein-protein interaction between chains, it is hard to accurately predict the structure of the domain and its orientation. For T1173 (Fig. 4c), the GDT-TS of the model

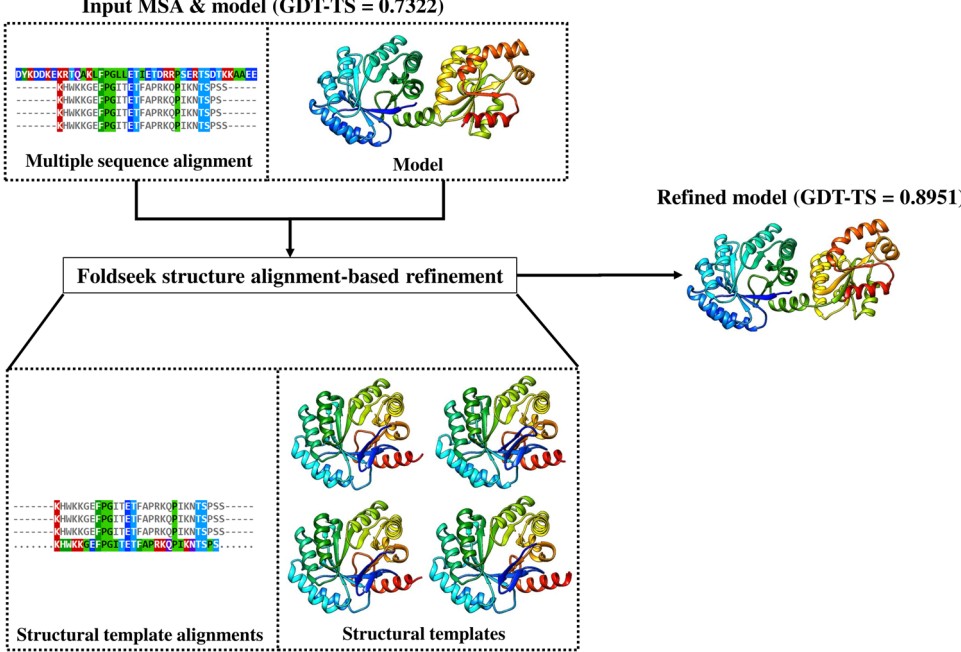

**Fig. 3 An example (T1180) of the Foldseek structure alignment-based refinement.** The illustration of the Foldseek structure alignment-based refinement on T1180, where the quality of the input model has been significantly improved (GDT-TS from 0.7322 to 0.8951).

extracted from the assembly structure predicted by the assembly structure prediction is 0.9203, much higher than 0.5502 of the single-chain structure prediction. The reason is that T1173 is one of the three identical chains of a homo-trimer T1173o, in which the three chains tightly interact with each other. Without the interaction partner information, it is impossible for the single-chain structure prediction to accurately predict the structure of a domain of the target and its orientation.

**Relationship between MSA and model quality**. It is known that the quality (e.g., the depth) of multiple sequence alignments has an impact on the quality of predicted tertiary structures. Figure 5A plots the average GDT-TS of the models generated by the default AlphaFold2 in the MULTICOM system against the logarithm of the Number of effective sequences (Neff) of the input MSA for the 91 domains from the 65 full-length targets (excluding T1114s1, T1114s2, and T1114s3) in Fig. 5A. The correlation between the logarithm of Neff and the GDT-TS is 0.4906, which is weak. However, if the domains belonging to an assembly target are excluded, the correlation between the two increases to 0.8374 (Fig. 5B), which is a much stronger correlation (Fig. 5B). Moreover, on the domains that are a part of the assembly targets, the correlation between the GDT-TS and the logarithm of the Neff is −0.009. The results clearly show that the depth of MSAs can explain the quality of the tertiary structures predicted for single-chain monomer targets, but it has little correlation with the quality of the tertiary structures predicted for multiple chains in a protein assembly where the protein-protein interaction plays an important role in shaping the tertiary structures of the chains.

Among the 91 domains, T1122-D1 and T1131-D1 have no other homologous sequences in their MSAs, i.e., their MSAs contain only one sequence (i.e., itself). The average GDT-TS of the tertiary structural models generated for the two domains is lower than 0.25, i.e., there is no good model among the dozens of models that AlphaFold2 generated for the two targets from their single-sequence MSA input. Before some special algorithm is developed to address this challenge of predicting protein structure

from a single sequence, it might be useful to run AlphaFold2 with different parameters to sample many more models, hoping some good models may be generated occasionally.

**Predicting structures for large multi-domain proteins**. For some large multi-domain proteins, it was time-consuming and computationally expensive to build the structure using Alpha-Fold2. Therefore, domain segmentation was applied to build the structure for very long sequences during the CASP15. For instance, for T1169 consisting of 3364 residues, we first built a model using the template-based structure prediction model using an early version of MULTICOM[21]. The template alignments showed that the first 350 residues and the last 614 residues could not be aligned to any templates and therefore were not folded in the structure. The first 350 residues and the last 614 residues were fed for the single-chain structure prediction to generate models for them separately. The full-length models for the target were also generated by AlphaFold2. Then, the poorly folded first 350 residues and last 614 residues in the top-ranked full-length models were replaced by the top-ranked models generated for each region separately to produce the final full-length models for T1169.

**Comparison between MULTICOM server predictors and human predictors**. The MULTICOM human predictors (MUL-TICOM and MULTICOM_human) used all the models generated by the MULTICOM servers and some additional models for some targets generated between the server prediction deadline and the human prediction deadline, particularly for the targets belonging to large protein assemblies. For some targets belonging to protein assemblies, more assembly structures were generated to extract more tertiary structural models for them. For instance, for targets T1137s1 and T1137s2 belonging to a protein complex (e.g., H1137), the server predictors used the sequence of a single chain (T1137s1 or T1137s2) to predict its tertiary structures because the sequences of other chains were not available when the server prediction was made, while the human predictors were able to predict the quaternary structures of the protein assembly first and

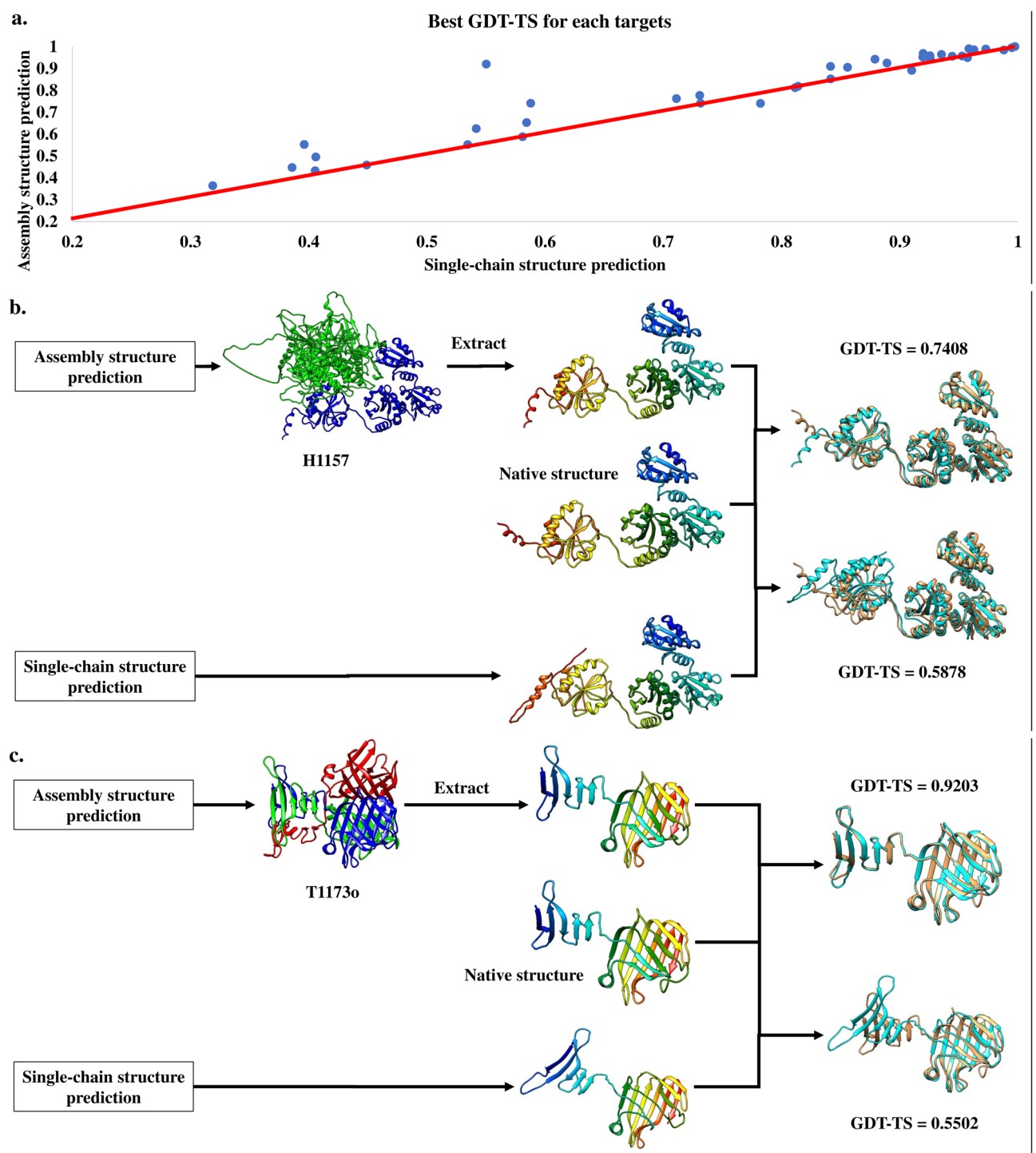

**Fig. 4 The comparison between the single-chain structure prediction and the assembly structure-based prediction. a** The plot of the GDT-TS of the best tertiary structural model generated by the assembly structure prediction (y-axis) for each of the 38 targets against that of the best model generated by the single-chain structure prediction. The assembly structure prediction generated better models than the single-chain structure prediction for 36 out of 38 targets (dots above the red line). In some cases, the increase of the GDT-TS score is substantial. The comparison between the best model generated by the single-chain structure prediction and the best model extracted from the assembly structure generated by the assembly structure prediction for (**b**) T1157s2 (a chain of heteromultimer H1157) and (**c**) T1173 (a chain of a homomultimer T1173o).

then extracted tertiary structures of the single chain from them as the tertiary structure predictions. This significantly improved the quality of the tertiary structure models predicted for these targets.

Supplementary Table 8 reports the average GDT-TS of top 1 models for all 94 domains, 39 domains from the single-chain targets, and 55 domains from the targets belonging to protein assemblies predicted by our two human predictors (MULTICOM, MULTICOM_human) and four server predictors (MULTICOM_egnn, MULTICOM_deep, MULTICOM_refine, MULTICOM_qa). Compared with the best MULTICOM server predictor - MULTICOM_refine, the best MULTICOM human predictor - MULTICOM performed slightly better on the 94

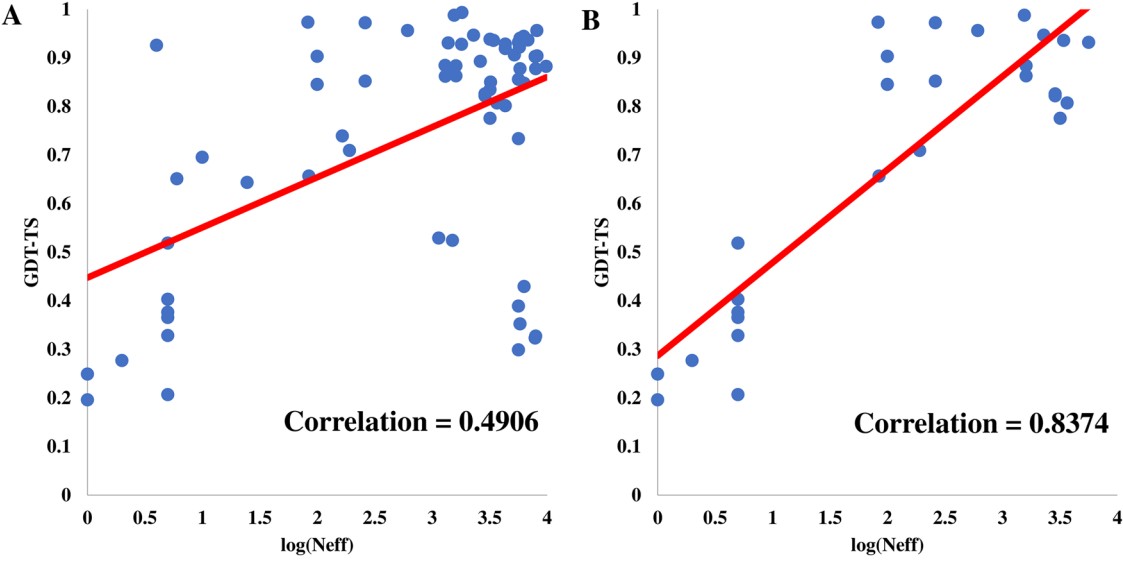

**Fig. 5 The logarithm of the Neff plotted against the average GDT-TS of the structural models. A** On 91 domains from 65 full-length targets. **B** On 39 domains from 27 single-chain full length targets.

domains in terms of the average GDT-TS score (0.8073 versus 0.7964) with no significant difference. They performed very similarly on the 39 domains from 27 single-chain targets (0.7520 vs 0.7538). The reason is that the size of the model pool of MULTICOM_refine is similar to that of MULTICOM. MULTICOM somewhat outperformed MULTICOM_refine on the 55 domains from 41 targets belonging to protein assemblies in terms of the average GDT-TS (0.8464 versus 0.8266), and the difference is significant (p-value = 0.02934 according to one-sided Wilcoxon signed rank test). One reason is that MULTICOM generated more assembly models for the assembly targets (e.g., H1137) to extract tertiary structures for the targets belonging to them. Particularly, for some targets such as T1137s1 and T1137s2, MULTICOM was able to extract the tertiary structures for them from the predicted assembly structures (e.g., H1137), but MULTICOM_refine only could use the single-chain structure prediction to generate structures for them, causing a big difference in the quality of their models.

**Comparison of quality assessment methods of ranking structural models.** We compare the three quality assessment (QA) methods used to select models for each target by the MULTICOM server predictors and the MULTICOM human predictors, including pLDDT score generated by AlphaFold2, average pairwise similarity score between a model and all other models calculated by APOLLO, and the average of the two scores (APOLLO$_{pLDDT\_avg}$). The performance is measured by two metrics: the average per-target ranking loss (i.e., per-target ranking loss = the GDT-TS score of the best model - the GDT-TS score of the top 1 selected model) and the average per-target correlation on all the targets (i.e., the per-target correlation for a target = the Pearson's correlation between the predicted quality scores and real GDT-TS scores of the models of the target). We compare the three QA methods on the structural models for these targets generated before the server prediction deadline (called server_model_dataset) and on all the structural models for the targets generated before the human prediction deadline (called human_model_dataset) respectively to investigate how their performance may change as the data set is changed. The human_model_data_set contains the sever_model_data_set and some additional models generated for some targets.

In addition to APOLLO$_{pLDDT\_avg}$, a simple way to combine pLDDT score and APOLLO score, another way is to use AlphaFold2 pLDDT score to calculate the weighted average pairwise similarity score between a model and all other models of a target, denoted as APOLLO$_{pLDDT\_weight}$. The weighted pairwise similarity score for model $j$ is $\frac{\sum_{i \neq j}^{n} Sim^{ij} * pLDDT^i}{n-1}$, where $pLDDT^i$ is the pLDDT score of any other model $i$, $Sim^{ij}$ is the structural similarity score (i.e.,TM-score) between model $i$ and $j$, and $n$ is the number of the models for a target.

The performance of the five QA methods on the server_model_dataset and the human_model_dataset is reported in Fig. 6A–F and Fig. 6G–L respectively. It is worth noting that the models in both the server_model_dataset and the human_model_dataset were all internally predicted by our in-house MULTICOM system without including any models predicted by the third-party predictors. On the server_model_dataset (Fig. 6A–F), AlphaFold2 pLDDT score selects models with the highest average GDT-TS of 0.7517 on the 68 targets (Fig. 6A) and the highest average GDT-TS of 0.7598 on the 27 single-chain targets (Fig. 6C), while APOLLO$_{pLDDT\_weight}$ has the highest correlation of 0.3880 on the 68 targets (Fig. 6B). However, on 41 targets whose models include both ones extracted from assembly models and the ones predicted by the single-chain structure prediction, APOLLO$_{pLDDT\_avg}$ has the highest average GDT-TS of 0.7483 (Fig. 6E), while APOLLO$_{pLDDT\_weight}$ has the highest correlation of 0.4690 (Fig. 6F). The results show that AlphaFold2 pLDDT, APOLLO and APOLLO$_{pLDDT\_avg}$ are complementary and their relative performance depends on the structural models to be evaluated to some degree. However, it is still difficult to combine APOLLO and AlphaFold2 pLDDT scores to obtain consistently better results.

On the human_model_dataset (Fig. 6G–L), APOLLO$_{pLDDT\_weight}$ has the highest average GDT-TS of 0.7511 on the 68 targets (Fig. 6G), while APOLLO has the highest correlation of 0.4470 on them (Fig. 6H). On 27 single-chain targets, AlphaFold2 pLDDT has the highest average GDT-TS of 0.7542 (Fig. 6I), while APOLLO$_{pLDDT\_avg}$ has the highest correlation of 0.4490 (Fig. 6J). On the 41 targets that are chains of protein assemblies, APOLLO$_{pLDDT\_weight}$ has the highest average GDT-TS of 0.7551 (Fig. 6K) and APOLLO has the highest correlation of 0.5051 (Fig. 6L). Overall, APOLLO$_{pLDDT\_weight}$ has higher average GDT-TS with AlphaFold2

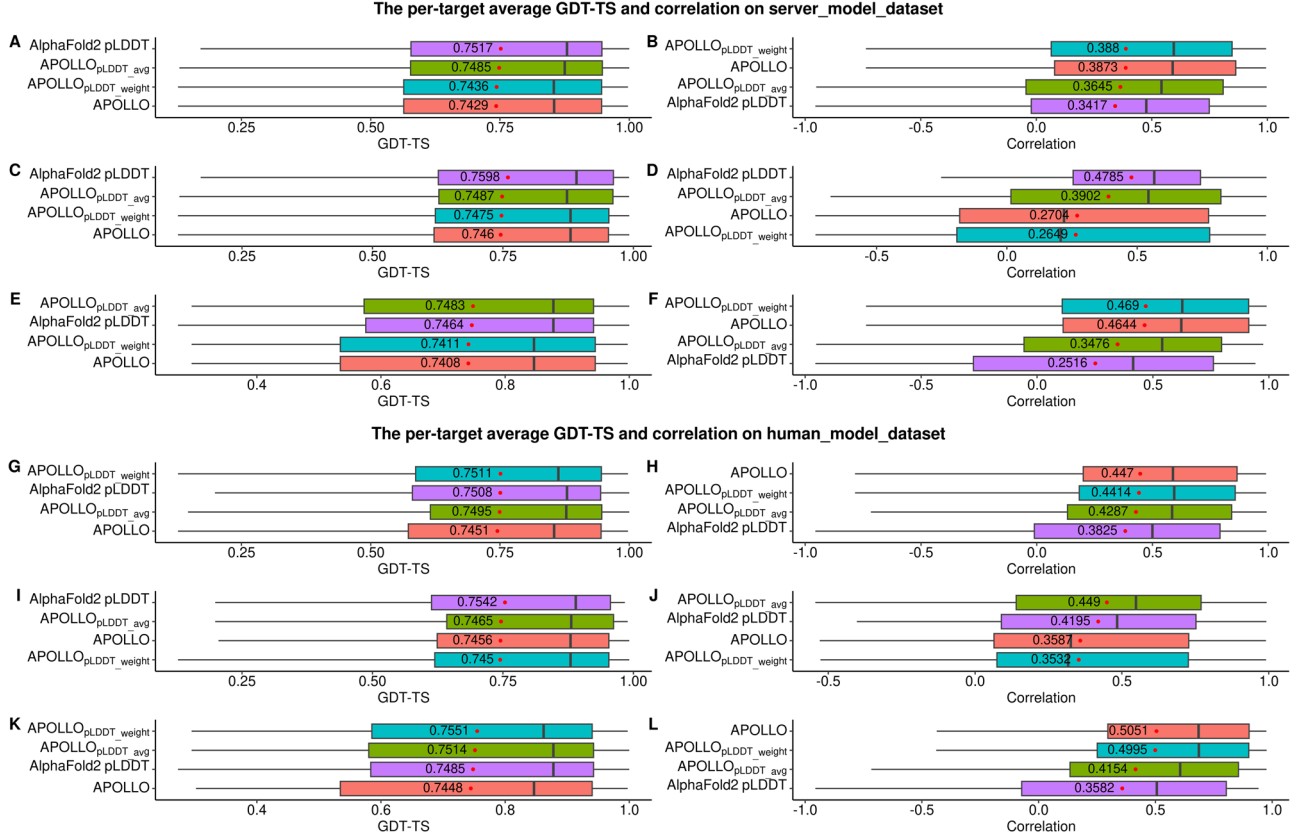

**Fig. 6 The performance of the four quality assessment methods on all 68 full-length targets, 27 single-chain full-length targets, and 41 full-length targets that are chains of the assembly targets, respectively. A** Per-target GDT-TS and **B** per-target correlation on the server_model_dataset of 68 full-length targets, where the average of per-target GDT-TS or correlation is shown in the box. **C** Per-target GDT-TS and **D** per-target correlation on the server_model_dataset of 27 single-chain full-length targets. **E** Per-target GDT-TS and **F** per-target correlation on the server_model_dataset of 41 full-length targets. **G** Per-target GDT-TS and **H** per-target correlation on the human_model_dataset of 68 full-length targets. **I** Per-target GDT-TS and **J** per-target correlation on the human_model_dataset of 27 single-chain full-length targets. **K** Per-target GDT-TS and **L** per-target correlation on the human_model_dataset of 41 full-length targets. The vertical line in each box denotes the median per-target value. The red dot in each box denotes the mean per-target value.

pLDDT (0.7511 vs 0.7508) and higher correlation (0.4414 vs 0.3825) on the 68 full-length targets. It also has a lower ranking loss (0.0384) on the 41 targets belonging to assemblies than AlphaFold2 pLDDT, APOLLO and APOLLO$_{pLDDT\_avg}$, while the correlation of APOLLO$_{pLDDT\_weight}$ (0.4995) is only lower than APOLLO (0.5051). In Supplementary Fig. 2a, the ranking loss of APOLLO$_{pLDDT\_weight}$ is plotted against that of AlphaFold2 pLDDT for the 68 full-length targets, which shows that APOLLO$_{pLDDT\_weight}$ performed better on the 41 targets from assemblies but worse on the 27 single-chain targets.

Finally, it is worth noting that AlphaFold2 pLDDT score is a single-model quality assessment (QA) method that evaluates the quality of a model without using any other model as input, while the other three QA methods use the pairwise similarity between models for model evaluation and therefore are multi-model QA methods. Therefore, AlphaFold2 pLDDT score does not depend on the diversity of the models in the model pool, while the multi-model QA methods requires that the model pool contains a sufficient number of relatively good models so that they may have higher pairwise similarity scores than the bad models as shown in Supplementary Fig. 2b. For T1180, there are more models with lower qualities (0.65~0.75), which makes APOLLO methods (e.g., APOLLO$_{pLDDT\_weight}$) difficult to select the best model. For T1106s1 which is a chain of H1106, most of the models have GDT-TS close to 0.85, and therefore APOLLO methods can easily select a good model. In contrast, AlphaFold2 pLDDT score

performs well on T1180 but not quite well on T1106s1 whose models include both the ones predicted by the single-chain structure prediction with AlphaFold2 and the assembly structure prediction with AlphaFold-Multimer. One possible reason may be that the pLDDT scores for the two kinds of models generated by AlphaFold2 and AlphaFold-Multimer were generated by the two different predictors and therefore were not completely comparable.

## Discussion

Our CASP15 results show that the MULTICOM system built on top of AlphaFold2 and AlphaFold-Multimer can predict high-accuracy protein tertiary structure for most targets and significantly improve the quality of structure prediction over the standard AlphaFold2 tool. One main reason for improvement is to use the protein assembly (complex) structure prediction to generate tertiary structure prediction for protein targets that are subunits of protein assemblies/complexes, considering the impact of protein-protein interaction on protein folding. Failing to consider protein-protein interaction for the non-globular protein targets (e.g., T1137s1, T1137s2, and T1137s3) that intertwine with their partners can lead to poor structure predictions.

Another main reason for improvement is to adjust the input (i.e., multiple sequence alignments and structural templates) to sample more structural models. Our CASP15 experiment demonstrates that using diverse MSAs and templates as input can

significantly improve the quality of the best models generated by AlphaFold2. The quality of the models for single-chain protein targets has a strong correlation with the depth of MSAs, but the quality of the models for protein targets that are subunits of protein assemblies has little correlation with the depth of MSAs, indicating that other factors such as protein-protein interaction play a more important role in determining the quality of models for such targets.

Even for some very hard FM targets with shallow MSAs (e.g., T1122), the single-chain structure prediction based on Alpha-Fold2 in MULTICOM still can generate some good models with correct topology (TM-score > 0.5), although the chance of generating a correct model for such a target is much lower. Supplementary Table 9 reports the fraction of models with correct topology for each CASP15 full-length target. The fraction of models with correct topology for T1122 is 2%, while all the models for all TBM targets except two outliers (T1160 and T1161) with alternative conformations have correct topology. For most FM and FM/TBM targets, more than 50% of their models have correct topology. But some hard targets, mostly the ones whose MSAs only have a few effective sequences or even a single sequence (e.g., T1130, T1131), have no model of correct topology among about dozens of models generated for them, indicating predicting tertiary structures from very shallow MSAs or a single protein sequence is still a major challenge. Generally speaking, harder a target is, the lower the probability of generating a model with correct topology for it. Therefore, it is expected that sampling more models for a hard target is necessary to obtain some high-quality models for such a target. In this case, other sampling strategies not explored in this work (e.g., predicting the structures only from MSAs without using templates, using the mono-mer_ptm network of AlphaFold2 instead of the monomer network, and adjusting the dropout rate of the AlphaFold2 tested by some top CASP15 predictors such as the Wallner group) can be applied to generate more models too. In fact, we performed post-CASP15 experiments on CASP15 targets using two different sampling strategies (i.e., predicting structures only from MSAs without using templates and using the monomer_ptm network of AlphaFold2 instead of the monomer network). Although there is no significant difference between each of these two sampling methods and the default setting of AlphaFold2 on average, for some targets, they can generate better models than the default setting. For example, for target T1119, AlphaFold2 can predict structure with 0.9619 TM-score without using any templates or generate a model with 0.9578 TM-score using the monomer_ptm network instead of the monomer network, both of which are higher than 0.8657 of the top1 model generated by using the default setting of AlphaFold2. Combining these sampling methods with our MSA and template diversity-based sampling methods to generate more models (e.g., hundreds or thousands) may further improve prediction accuracy. Therefore, it would be interesting to investigate how many models need to be generated in order to sample a correct structural model for a hard protein target in the future.

It is also useful to investigate if AlphaFold2 can generate a correct structural model for any protein target no matter how hard it is and how shallow MSA is, provided that a sufficient number of structural models is generated. If the answer to this question is yes, then the problem of sampling protein structures can be considered fully solved.

As AlphaFold2 can generate some good structural models for most if not all protein targets if it samples a sufficient number of structural models, the next significant challenge is to select good structural models from a large number of models generated by AlphaFold2. The selection problem can be difficult for hard targets for which only a small portion of models are of good quality.

As shown in our CASP15 experiment, although both the pLDDT score predicted by AlphaFold2 and other model quality assessment methods tested (e.g., the average pairwise similarity score (PSS) between a model and other models) can do a reasonable job, they still cannot select very good models for some targets. The AlphaFold2 pLDDT score is complementary with PSS, but it is still challenging to combine them to consistently obtain better results in evaluating the quality of protein tertiary structural models. Quite some model quality assessment methods had been developed before AlphaFold2 was made available, but they were trained to evaluate the quality of models generated by pre-AlphaFold2 structure predictors and might not perform well with AlphaFold2 models. Therefore, a new generation of model quality assessment methods that can improve the estimation of the accuracy of AlphaFold2-generated models needs to be developed.

Once a structural model is selected for a target, there is a possibility to further improve its quality through model refinement. Our CASP15 experiment demonstrates that the Foldseek structure alignment-based approach of augmenting MSAs and structural templates can help AlphaFold2 to generate some models better than the initial input model. In some cases, the improvement is substantial, even though the approach does not significantly improve model quality substantially on average. The results suggest that it may be useful to further explore the direction of using protein structure search and alignments to improve AlphaFold2-based protein structure prediction and sampling.

Finally, it is still challenging to sample protein tertiary structures for very large protein targets that have thousands of residues. First, sampling structural models for such targets requires a lot of GPU memory that is not readily available and takes a lot of time. Second, the MSAs generated for the entire target sequence could be shallow (e.g., only 5 homologous sequences for T1125). Third, the MSAs generated for them may not cover the entire target sequence well. Particularly they can be shallow for some domains of the target (e.g., the first 350 residues and last 614 residues of T1169). In this case, in addition to sampling full-length models, it is useful to divide the target into multiple domains/regions to predict the structures of individual domains/regions separately and then combine them with the full-length models.

## Conclusion

We developed several methods to improve the process of using AlphaFold2 to generate protein structures and blindly tested them in the CASP15. We demonstrate that generating diverse MSAs and structural templates using different alignment protocols and protein databases can improve the quality protein structural models. It is also critical to use the protein assembly structure prediction to predict the tertiary structures for targets that are subunits of protein assemblies in order to account for the impact of protein-protein interaction on protein folding. More-over, an iterative structure-alignment based approach of generating MSAs and identifying templates can further refine some protein structures predicted by AlphaFold2 substantially.

Interestingly, our MULTICOM system that only generated a small number of protein structural models for CASP15 targets on average was able to sample at least one model with correct topology for the vast majority of targets from diverse alignments and structural templates, even though the probability of generating good models depends on the quality of the input (e.g., MSA) and difficulty of the target. This raises an interesting question if AlphaFold2 can generate a good structural model for any target of any difficulty provided a sufficient number of models are simulated, and if so, how many simulations are

**Fig. 7 The overall workflow for the MULTICOM protein tertiary structure prediction system.** The single-chain structure prediction process consists of five sequential steps: (1) multiple sequence alignment sampling, (2) template identification, (3) monomer structural model generation, (4) structural model ranking, and (5) Foldseek-based iterative model refinement. If the target is a chain of a protein assembly, the target and other chains in the protein assembly are also fed into the assembly structure prediction module built on top of AlphaFold-Multimer to generate quaternary structures. The tertiary structures of the target are then extracted from the predicted quaternary structures and are added to the structural model pool generated by the single-chain protein structure prediction.

needed for a target. Given AlphaFold2's powerful capability of sampling protein tertiary structures, we also investigated different methods of ranking AlphaFold2 structural models. Our experiments show that AlphaFold2 pLDDT score and the pairwise similarity score perform reasonably well and are complementary. As the AlphaFold2-based model sampling reaches a very high level, it is imperative to develop more effective methods to rank protein tertiary structural models generated by AlphaFold2.

## Methods
**Overview of the MULTICOM tertiary structure prediction system**. Figure 7 illustrates the overall workflow of the MULTICOM tertiary structure prediction system, which is a combination of the single-chain structure prediction and the assembly structure prediction. As shown in Fig. 7, the sequence of an input monomer target is always fed into the single-chain structure prediction module to generate structural models. The single-chain structure prediction process consists of five sequential steps: (1) multiple sequence alignment sampling, (2) template identification, (3) monomer structural model generation, (4) structural model ranking, and (5) Foldseek-based iterative model refinement. Except for Step 3 (monomer structural model generation) that is handled by the pre-trained deep learning models of

AlphaFold2, all the other steps are largely based on our customized algorithms. If the target is a chain of a protein assembly, the target and other chains in the protein assembly are also fed into the assembly structure prediction module built on top of AlphaFold-Multimer[22] to generate quaternary structures. The tertiary structures of the target are then extracted from the predicted quaternary structures and are added to the structural model pool generated by the single-chain protein structure prediction. The details of the algorithms are described in the next few sections.

**Single-chain tertiary structure prediction**.

(1) Monomer multiple sequence alignment (MSA) sampling. Given the sequence of a protein target, different kinds of MSAs are sampled from various sequence databases including UniRef30[23] (UniRef30_2021_02), UniRef90[24] (version 04/24/2022), BFD[25, 26], MGnify clusters[27] (mgy_clusters_2022_05) and the ColabFold DB[28]. HHblits[29] is applied to search homologous sequences on UniRef30, UniRef90 and BFD using parameters -n = 3, -e = 0.001, -maxseq=1_000_000, -realign_max=100_000, -maxfilt=100_000, -min_prefilter_-hits=1000. JackHMMER[30] is applied for searching UniRef90

using parameters --F1 = 0.0005, --F2 = 0.0005, --F3 = 0.0000005, --incE=0.0001, -E = 0.0001, -N = 1. MMseqs2[31] is used to search ColabFold DB. Moreover, a DeepMSA[32]-like alignment tool is executed in the background to iteratively search the UniRef90, huge Integrated Microbial Genomes (IMG) database[33] and the metagenome sequence databases (e.g., BFD, Metaclust[26], MGnify clusters) to generate alternative alignments for hard targets having few homologous sequences (e.g., < 200). The combination of different sequence search tools and sequence databases yields different kinds of MSAs for a target (Table 3). Only default MSA and default_seq_temp are generated by the default program of the original AlphaFold2 from the updated version of the same protein sequence databases used by the default AlphaFold2. The other MSAs are generated by different sequence search tools or from different sequence databases.

(2) Template identification. Similar to the template searching process in AlphaFold2, the structural templates are identified by searching the sequence profile of a target built from its MSA against a template database (pdb70 or PDB_sort90) using HHsearch[34]. The MSA of a target generated from UniRef90 is used as input for HHsearch to search pdb70 (version 3/13/2022) in the AlphaFold2 package and our inhouse template database (PDB_sort90) curated from Protein Data Bank[11] (PDB) to identify alternative templates. PDB_sort90 was constructed by several steps as follows: Firstly, all the structures in both pdb and mmcif format were downloaded from PDB. Secondly, for the same PDB code, only one structure file in pdb or mmcif format was kept. Thirdly, the structure of each chain was extracted from every structure file. Fourthly, the structures that have a resolution > 8 angstrom, less than 30 residues, or more than 90% sequence identity with other proteins were filtered out. Fifthly, the MSA for each remaining protein was generated by using HHblits to search it against the UniRef30_2021_03 database. Finally, the MSAs were used by the ffindex_build tool in the HHsuite-3.2.0[35] package to create the PDB_sort90 template database. The template database is used to identify templates for a target by using HHsearch to search its MSA against the template database (Table 3).

(3) Monomer structural model generation. To leverage the power of AlphaFold2, a customized version of AlphaFold2 that accepts pre-generated MSA and templates is built and used to generate models. Different from the default parameters of the original AlphaFold2, the value of parameter num_ensemble is changed from 1 to 8 and num_cycle from 3 to 8 to perform more extensive model sampling. The customized AlphaFold2 takes each pair of MSA and its corresponding templates in Table 3 that have been generated for a target as input to generate 5 structural models. Multiple combinations of MSAs and templates lead to up to 40 models generated for each target, depending on the number of MSAs generated for the target. If the depth of the default MSA is larger than 200, two MSAs (img and img_seq_temp) will not be used to generate models.

(4) Structural model ranking. The APOLLO[12] model ranking score (the average pairwise structural similarity between a model and the other models of the same target) and the global pLDDT score generated by AlphaFold2 are used to rank the structural models, respectively. The average of the two is also used to rank them. Moreover, a deep learning method - DeepRank[36] is used in model ranking. An early version of EnQA[37] – a 3D-equivariant deep learning model is applied to rank the structural models when appropriate.

(5) Foldseek structure alignment-based refinement. We developed a novel iterative model refinement method based on Foldseek[15] structure alignment (comparison) method (Supplementary Fig. 3). An initial structural model is used as input for Foldseek to search for similar structures in the PDB_sort90 template database and the AlphaFoldDB (the version released before March 2022). The output of the Foldseek includes the e-value (and TM-score[38] if TMalign[39] option is used) of the similar structural hits as well as the structural alignments between the target model and the hits, which are converted into the sequence alignments between them. The sequence alignments are added into the original MSA used to generate the initial structural model to generate a deeper MSA. The redundant sequences in the new MSA are removed by HHfliter[4] according to the 90% sequence identity threshold. The filtered MSA and the top-ranked structural hits found by Foldseek are used as MSA and template inputs for the customized AlphaFold2 to generate the refined models. If the highest AlphaFold2 pLDDT score of the newly refined models is higher than that of the input model, the refinement process is repeated with the refined model with the highest pLDDT sore as input until the number of the refinement iterations reaches 5 or the pLDDT does not increase anymore. The refined model with the highest pLDDT score generated in the refinement process is used as the output model.

**Prediction of tertiary structures of proteins involving in protein-protein interaction by integration of monomer and assembly structure prediction.** If an input monomer target is a chain of a protein assembly interacting with other chains, the assembly structure prediction module[13] built on top of AlphaFold-Multimer is applied to generate structures for the protein assembly first. After collecting the structural models with the pickle files for the protein assembly generated by AlphaFold-Multimer, the tertiary structure of the target and the local pLDDT scores of its residues are then extracted from the quaternary structure and its corresponding pickle file and added into the structural model pool for the target. Different from predicting the tertiary structure of the target from its sequence alone, this approach considers the interaction between the target and other chains and therefore can predict the changes on tertiary structures induced by the protein-protein interaction.

**Implementation of MULTICOM server and human predictors in CASP15.** The MULTICOM system was used to generate protein structural models for the monomer targets in the CASP15 experiment for the server prediction before the server prediction deadline. After the server prediction deadline and before the human prediction deadline, it continued to generate some additional models for hard targets or large targets if necessary. The models were ranked by different quality assessment methods, leading to the four different MULTICOM server predictors and two human predictors as follows.

If a monomer target is not a part of a protein assembly, MULTICOM_egnn server predictor used the average of the pLDDT score and the APOLLO pairwise similarity score to rank models. MULTICOM_refine refined the top five models selected by the average ranking and selected the final five models with the highest pLDDT scores from the 5 unrefined models and 5 refined models. MULTICOM_deep used pLDDT score to rank and select models. MULTICOM_qa refined the top 5 models generated by the AlphaFold2 with the default MSA in Table 3. The two human predictors (MULTICOM and MULTICOM_human) selected

monomer models from a generally larger model pool than the server predictors. The refined models were also added into the pool for ranking. DeepRank[13] was used to rank models for MULTICOM_human, while the average ranking of the pairwise similarity score and AlphaFold2 pLDDT score was used to rank models for MULTICOM. The ranking may be manually adjusted according to the human inspection.

If the monomer target is a chain of a protein assembly, the tertiary structural models for the target extracted from the assembly models were preferred to the structural models generated by the single-chain structure prediction without considering the interaction between chains. Generally, the top ranked models extracted from assembly models were used as the top 3-4 models submitted to CASP15, while the remaining models submitted could be the top ranked models generated by the single-chain tertiary structure prediction.

## Data availability

The protein structures of CASP15 monomer targets are available at https://predictioncenter.org/download_area/CASP15/targets/. The protein structural models and analytical data generated in this study are available at https://doi.org/10.5281/zenodo.8215939. All other data are available from the corresponding author on reasonable request.

## Code availability

The source code of MULTICOM add-on package for AlphaFold2 are available at: https://github.com/BioinfoMachineLearning/MULTICOM3.

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

## Acknowledgements

We thank CASP15 organizers and assessors for making the CASP15 data available. We also thank Dr. Soeding's group for releasing the latest HHsuite protein hidden Markov model (HMM) database for the community to use prior to the CASP15 experiment. This work is partially supported by two NIH grants [R01GM093123 and R01GM146340], Department of Energy grants [DE-SC0020400 and DE-SC0021303], and three NSF grants [DBI1759934, DBI2308699, and IIS1763246].

## Author contributions

J.C. conceived the project. J.C. and J.L. designed the experiment. J.L., J.C., Z.G., T.W., R.R., and C.C. performed the experiment and collected the data. J.L. and J.C. analyzed the data. J.L. and J.C. wrote the manuscript.

## Competing interests

The authors declare no completing interests.
