## [Peer Review File · Communications Chemistry]

Reviewers' comments:

Reviewer #1 (Remarks to the Author):

This paper summarizes the strong performance of MULTICOM in the recent CASP competition. The methodological contribution is a pipeline that first improves the depth of multiple sequence alignment (MSA) with a new MULTICOM method, the input needed for AlphaFold2, then utilizes AlphaFold2 to generate top models, and then further refines those models with MULTICOM-refine. Though the whole system can be identified as MULTICOM, the name itself refers to the part that improves the MSA.

The authors convincingly show that the performance of this pipeline is strong, provide rankings, GDT-TS improvements over other methods and the baseline AlphaFold on TBM and FM targets, monomeric and multimeric.

They convincingly show that improving the quality of the MSA (known as depth) is the single most important knob to improving the performance of AlphaFold2 and hence the quality of generated and final (after refinement)_structures.

This per se, though very useful to see in action, is not a new observation. We have long known that the quality of structure prediction depends on the quality of the MSA. As the authors point out, on sequences with no homologous sequences, there is not much that can be done except to go to ESM2-Fold, which, while not as accurate as AlphaFold2, can at least provide a prediction.

While I appreciate the thorough work and the contribution of the authors, I am struggling to resolve to myself the question of whether these findings rise to a Communications in Chemistry report in Nature. This is a great engineering exercise, but ultimately it does not provide us with any foundational, novel insight. It improves quality by a few points, but, ultimately, it is hard to appreciate the scientific contribution.

Let me put this another way. The paper essentially does the following: we take good and make it better. This is great, but I am not convinced this is sufficient for a nature publication. What would have been more interesting and rising to the level of a nature publication is the following: we take cases where you cannot build an MSA and give you a workable, accurate model. That is a Nature paper. What is presented in this paper is effectively a CASP summary report.

Reviewer #2 (Remarks to the Author):

The manuscript from Cheng group describes the performance of their MULTICOM suite in CASP15 tertiary structure prediction category. The method is running AlphaFold with MSAs constructed against different databases, and templates against different template databases, followed by a quality assessment of the pool of models to select the best model. Overall, the method was ranked third among all servers in CASP15.

The manuscript proposes an important claim: the potential to enhance AlphaFold2 predictions through additional sampling. However, the current writing style resembles a CASP15 method paper, which limits its appeal beyond the CASP15 participants. To reach a wider audience, I recommend reformatting the manuscript.

While the manuscript raises intriguing questions, it predominantly focuses on reporting the method employed in CASP15, rather than addressing these questions directly. To augment the manuscript's impact, I suggest conducting post-CASP15 analyses to evaluate the effectiveness of the selected parameters and settings. Additionally, it would be valuable to explore the feasibility of the proposed "easy" suggestions, such as those instances where the authors mention the potential for generating improved results (e.g. all instances of "may be able to generate better..."). Verifying these claims through simple checks would significantly strengthen the manuscript's credibility.

The initial introduction of the various settings and sampling methods could benefit from improvement, as they currently resemble internal variable names rather than clear explanations. To enhance the clarity and understanding for readers, I recommend refining the descriptions of these settings. Considering the importance of Table S7 for comprehending the different settings, including it in the main text would be advantageous. This inclusion would ensure that readers have access to the necessary information when exploring the various settings discussed in the manuscript. Furthermore, it would be helpful to specify which network was utilized: model or model_ptm. Additionally, providing details about the original settings, such as num_recycles=3 and num_ensemble=1, could be valuable. Consider incorporating this information into Table S7. Lastly, it is worth considering whether additional diversity could have been achieved by utilizing both model and model_ptm? Exploring this possibility and discussing its potential impact on the results would further enrich the manuscript.

Figure 2 shows the best models generated by different sampling methods compared to the AlphaFold baseline. What does a similar plot for rank 1 look like? In addition, the combine method is clearly sampling more structures than the AlphaFold baseline; thus, using the best model is not a fair comparison in this case. A larger pool of models will have a higher chance of generating a better model. It should be clear how many models the maximum is selected from, and for a comparison against the baseline it can only be for the rank 1. However, another more interesting question than improving over the baseline is which settings are best, and that is not even tested. Most settings could be compared head-to-head on a common set of 58 (?) targets instead.

How different are the distributions for the different sampling techniques for each target, i.e, the combination of all settings seems to generate a diverse set of models, but are they only sampling "one" conformation internally?

What about not using any templates? Sometimes a wrong template can confuse the method and produce a wrong model. You should test: no-template, pdb70, PDB_sort90.

The combine method from which settings are the rank1 models originating?

The quality assessment should be analyzed using selected model quality for each target, like Figure 2 on

rank 1 (selected) models or scatterplots with correlations. Are the reported correlations overall or per-target?

Using consensus or comparisons of server/human models is very limited outside the CASP setting, so it would be better if the authors could discuss the performance of a QA that only uses internally generated models that can be reproduced outside the setting of CASP, or single-model QAs.

The authors claim they are using sampling, but the sampling is very minor as each setting only generated five models; thus, with 8 settings, 40 different models are generated. Some of the more successful groups in CASP15 generated thousands of models using dropout to improve the diversity of models. Is that something that could be used here as well?

A related question is whether the MSAs could be resampled, assuming they are large, instead of generating new ones against different databases. Could too many sequences be a problem?

Concerning the Foldseek structure alignment-based refinement. How did it actually perform? If the mean performance did not change, and some targets were significantly improved. It implies that some targets also got significantly worse (otherwise, the mean would also be higher). Either report the performance or remove it; since it was only used for 27 targets and the performance gain is weak, it does not need to be highlighted in the abstract.

For the T1180 case in Figure 3, is the four templates found by Foldseek more or less identical? And how similar are they to the one found by the sequence search? It would be clearer if the rainbow coloring is based on the target sequence, but it seems as if the templates only cover the N-terminal part. Would it have worked by just injecting the sequence-found template four times, effectively increasing the weight on the templates? Again, the paragraph has a “may be the reason...”; please check the reason. In addition, by comparing the new templates in Figure 3, it seems as if the model improvement is in the part that is not in the templates, or am I wrong? If that is the case, more sampling/recycles is more likely the reason for the improvement rather than a better template found by Foldseek. Please, comment on this.

Minor:

Starting paragraphs with boldface Figure #no is confusing, at least in manuscript form; they look almost like the caption.

Figure 2 caption says GDT, figure says TM.

Response to Review Comments

Reviewer 1:

Comments to the authors:

This paper summarizes the strong performance of MULTICOM in the recent CASP competition. The methodological contribution is a pipeline that first improves the depth of multiple sequence alignment (MSA) with a new MULTICOM method, the input needed for AlphaFold2, then utilizes AlphaFold2 to generate top models, and then further refines those models with MULTICOM-refine. Though the whole system can be identified as MULTICOM, the name itself refers to the part that improves the MSA.

The authors convincingly show that the performance of this pipeline is strong, provide rankings, GDT-TS improvements over other methods and the baseline AlphaFold on TBM and FM targets, monomeric and multimeric.

They convincingly show that improving the quality of the MSA (known as depth) is the single most important knob to improving the performance of AlphaFold2 and hence the quality of generated and final (after refinement)_structures.

This per se, though very useful to see in action, is not a new observation. We have long known that the quality of structure prediction depends on the quality of the MSA. As the authors point out, on sequences with no homologous sequences, there is not much that can be done except to go to ESM2-Fold, which, while not as accurate as AlphaFold2, can at least provide a prediction.

While I appreciate the thorough work and the contribution of the authors, I am struggling to resolve to myself the question of whether these findings rise to a Communications in Chemistry report in Nature. This is a great engineering exercise, but ultimately it does not provide us with any foundational, novel insight. It improves quality by a few points, but, ultimately, it is hard to appreciate the scientific contribution.

Let me put this another way. The paper essentially does the following: we take good and make it better. This is great, but I am not convinced this is sufficient for a nature publication. What would have been more interesting and rising to the level of a nature publication is the following: we take cases where you cannot build an MSA and give you a workable, accurate model. That is a Nature paper. What is presented in this paper is effectively a CASP summary report.

Response:

Thank you for the valuable critiques and the opportunity for us to clarify the scientific contribution of this work. Even though our manuscript presents the results of our MULTICOM methods in the CASP15 experiment, it is very different from a typical CASP15 predictor report because the manuscript includes several novel algorithms that have never been published before and can be

generally applicable to AlphaFold-based protein structure prediction as well as some valuable scientific findings that can advance protein structure prediction. Typical CASP summary papers usually report the results of the methods that have been previously published. However, in this work we introduced several new, unpublished algorithms or scientific findings below that improve the accuracy of protein structure prediction over the standard AlphaFold2 widely used in the field:

First, we introduced several specific sampling methods using AlphaFold2 with diverse multiple sequence alignment (MSA) and template inputs to generate protein structures, showing that increasing the diversity of MSAs and templates can significantly improve the quality of the best/top1 models predicted by standard AlphaFold2. The unique capability of our approach is that it only needs to sample dozens of structural models with AlphaFold2 for a target to get significantly better predictions than the standard AlphaFold, which is different from other top CASP15 predictors that sample thousands of structural models.

Second, we introduced the first, iterative structure alignment-based refinement method in the field to enhance the MSA to improve AlphaFold2-based protein structure prediction, which can drastically improve the quality of the structures predicted for some targets (e.g., the GDT-TS of the refined model for T1180 is 0.8951, much higher than 0.6834 of NBIS-AF2-standard – the standard AlphaFold2 method). This structure alignment-based approach is novel and totally different from the traditional sequence-alignment based approach used in the field. Therefore, this approach opens a new direction of using protein structure alignment to generate inputs for AlphaFold2 to improve protein structure prediction. We expect to see more such methods to be developed in the field soon. We think this conceptual innovation can be considered a significant scientific contribution.

Third, our research systematically proved that on 36 out of 38 targets that are the chains of protein assemblies, the best tertiary structural model generated by the assembly structure prediction has higher GDT-TS than the tertiary structural model generated by single-chain structure prediction. The result highlights the significance of considering protein-protein interactions when predicting the tertiary structure of a monomer that is a subunit of a protein assembly.

Fourth, we show that the protein model pairwise similarity can be used to rank protein structural models and is complementary with the AlphaFold2's pLDDT score. For some targets, it can select better models than AlphaFold2's pLDDT score. This scientific finding has not been reported before and provides a new approach and direction to improve the ranking of AlphaFold2 predicted protein structures, which is much needed in the field.

Finally, in this revision, we have conducted a series of new post-CASP15 experiments to validate the new findings elucidated in this research (see the changes highlighted red in the revised manuscript). We have also improved the writing style to emphasize the new methods in this revised manuscript to make it more different from typical CASP papers reporting the results of existing methods.

By integrating the innovative algorithms and new findings above, our system has shown the capability to generate significantly improved models compared to the widely used standard AlphaFold2 on the CASP15 targets (i.e., 9.6% improvement in terms of GDT-TS and 8.2% improvement in terms of TM-score). This improvement is substantial and valuable because AlphaFold2 is widely used in the field and an improvement like this can immediately be adopted by the community and have a significant scientific impact on applying AlphaFold2 to solve all kinds of biomedical research relying on predicted protein structures.

Reviewer 2:

Comments to the authors:

The manuscript from Cheng group describes the performance of their MULTICOM suite in CASP15 tertiary structure prediction category. The method is running AlphaFold with MSAs constructed against different databases, and templates against different template databases, followed by a quality assessment of the pool of models to select the best model. Overall, the method was ranked third among all servers in CASP15.

The manuscript proposes an important claim: the potential to enhance AlphaFold2 predictions through additional sampling. However, the current writing style resembles a CASP15 method paper, which limits its appeal beyond the CASP15 participants. To reach a wider audience, I recommend reformatting the manuscript.

1. While the manuscript raises intriguing questions, it predominantly focuses on reporting the method employed in CASP15, rather than addressing these questions directly. To augment the manuscript's impact, I suggest conducting post-CASP15 analyses to evaluate the effectiveness of the selected parameters and settings. Additionally, it would be valuable to explore the feasibility of the proposed "easy" suggestions, such as those instances where the authors mention the potential for generating improved results (e.g. all instances of "may be able to generate better..."). Verifying these claims through simple checks would significantly strengthen the manuscript's credibility.

Response:

Thanks for the great suggestions. We have performed additional post-CASP15 experiments to evaluate the effectiveness of selected settings and verify the important claims in the manuscript according to the specific comments in your review. Below is a list of major changes.

In section "**The importance of sampling more models using AlphaFold2 with diverse MSA and template inputs**", we added new **Figure 2B** of comparing top1 models of different methods ranked by AlphaFold2's pLDDT score. The average TM-score of the top1 models generated by the "*combine*" method for 62 common targets is 0.786, higher than 0.77 of NBIS-AF2-standard. In new **Figure 2C**, the TM-score difference between the top1 model of the "*combine*" method selected by AlphaFold2's pLDDT score and the top1 model of NBIS-AF2-standard as well as the

number of models of the “*combine*” method is reported for each target. On most targets, the difference is positive, i.e., the top1 model of the “*combine*” method has a higher TM-score than NBIS-AF2-standard. As shown in new **Figure 2B**, four sampling methods (i.e., *default*, *default_seq_temp*, *original* and *ori_seq_temp*) have higher average TM-score of top1 models than NBIS-AF2-standard on the common targets, but there is no significant difference between each of them and NBIS-AF2-standard. This new analysis quantifies the amount of the difference between our MSA and template sampling methods and the standard AlphaFold2 method – NBIS-AF2-standard.

To better compare the performance between the in-house sampling methods that generated models for most of the CASP15 targets, the average TM-scores of the top1 models from the six sampling methods are shown in new **Figure 2D**, ordered by their average TM-score of the top1 models. The results of the one-sided Wilcoxon test indicate that there is no significant difference between “*original*” and “*ori_seq_temp*”, “*original*” and “*default_seq_temp*” in terms of the TM-score of the top1 models. However, there is significant difference between “*original*” and each of the other three sampling methods (*default*, *colabfold*, *colab_seq_temp*), with the p-value of 0.04604, 0.01678 and 0.01846 respectively.

To quantify the distribution of the similarity of the models generated by the different sampling methods, the average pairwise similarity score (PSS) of the models produced by each method is calculated. A higher average PSS indicates that the models in the model pool of a method are more similar, while a lower PSS suggests the presence of multiple or more diverse conformations in the model pool. To visualize the results, the average PSS of the models from “*combine*”, “*default*”, “*default_seq_temp*”, “*original*”, “*ori_seq_temp*”, “*colabfold*” and “*colab_seq_temp*” are plotted for each of the 58 common targets in new **Figure 2E**. The figure shows that, for 33 out of 58 targets, the average PSS of the methods is greater than 0.9, indicating that they all generated models of similar/same conformations. However, for 22 out of 58 targets, the average PSS of the “*combine*” model pool is less than 0.8, suggesting there is a diverse set of models in the model pool. Interestingly, for T1104, T1179, T1119, T1123, T1178 and T1154, some methods generated very similar conformations (e.g., the average PSS of “*colabfold*” models generated for T1104 is 0.8751), while the other methods (e.g., “*default_seq_temp*”) generated models with more different conformations (the average PSS of “*default_seq_temp*” for T1104 is 0.6247). The diversity of the MSAs and templates used by the different methods increases the variety of models in the “*combine*” model pool for these targets. The average TM-score of top1 models of the “*combine*” method on the 33 targets that has PSS value greater than 0.9 is 0.8858, only slightly higher than 0.8823 of NBIS-AF2-standard, indicating that the targets are mostly easy and generating more models from diverse MSAs and templates only have a small effect on the performance. However, on the 22 targets for which the PSS value of top1 models of the “*combine*” method is less than 0.8, the average TM-score of the top1 models of the “*combine*” is 0.6344, notably higher than 0.6073 of NBIS-AF2-standard, indicating that these targets are mostly harder and generating more models from diverse MSAs and templates have a larger effect on the performance. Indeed, for 14 out of the 22 targets, the top1 model in the “*combine*” model pool has a higher TM-score than that of NBIS-AF2-standard, showing that the increased variety of models improves the quality of 63.64% of these targets. *This new analysis determines when our sampling methods*

improve model quality and confirms that the MSA and template diversity improve the quality of predicted structures mostly for harder targets.

In section “**The effect of Foldseek structure alignment-based refinement method**”, we added new **Table S7** (the average GDT-TS of the original models and that of the refined models) to show the performance of the refinement method in details.

To investigate the factors that caused the improvement on T1180, we performed the following post-CASP15 experiments with AlphaFold2 to generate 15 models respectively, which was the total number of models produced / used during the refinement process. The refinement process used the five models of the “*default*” sampling method as initial models to generate 5 models in each of the three refinement iterations, resulting in 15 refined models in total for selection.

In the first experiment, the “*default*” sampling method was used to generate 15 models for T1180. The GDT-TS of the top1 model is 0.8583, higher than 0.7322 of the top1 model among the five models initially generated by the “*default*” sampling method, but lower than 0.8951 of the final refined model from the refinement. The experiment indicates that increased sampling can yield models of higher quality, but still cannot reach the quality of the refinement process.

In the second experiment, the combined alignments in each iteration of the refinement process along with the sequence search-found templates were fed to AlphaFold2 to generate 15 models. The GDT-TS of the top1 models for each iteration is 0.666, 0.8661 and 0.8806. The final score of 0.8806 is very close to 0.8951 of the final refined model submitted to CASP15, demonstrating that iteratively adding Foldseek-found structure alignments into the MSA in the refinement process is a reason for the improvement in model quality.

In the third experiment, the four templates (3JS3A, 3JS3B, 3NNTB, 4H3DB) identified by Foldseek along with the initial MSA were provided to AlphaFold2 to generate 15 models. The GDT-TS of the top1 model of the refinement process is 0.71, much lower than 0.8951 of the final refined model submitted to CASP15, indicating that the structural templates is not the reason leading to the model quality improvement. Furthermore, we used one template 4H3DB four times as structural templates for AlphaFold2 to generate models, resulting in a top1 model with GDT-TS of 0.7231, much lower than 0.8951 of the final refined model submitted to CASP15. This further confirms that adding more templates into the AlphaFold2 model generation is not the reason that the refinement process produced the high-quality models in CASP15.

This new analysis not only corrects one incorrect claim in the previous version of the manuscript that the new templates found by Foldseek helped improve the model quality but also identify the two true factors leading to the improvement (i.e., the improved MSA by Foldseek-based structure alignment and the increased number of models generated).

Furthermore, in the **Discussion Section**, we have performed a further analysis using two other easy sampling strategies (i.e., predicting the structures only from MSAs without using templates and using the monomer_ptm network of AlphaFold2 instead of the monomer network) other than

adjusting the dropout rate of AlphaFold2 since it has been tested by other CASP15 groups. The result has been discussed in the Discussion section. This new analysis validates the value of these two parameter settings for AlphaFold2.

By conducting the new experiments and analyses above, we have been able to validate the valid claims and useful algorithmic settings and provide definite conclusions about the important findings of this work that users can adopt in their work.

2. The initial introduction of the various settings and sampling methods could benefit from improvement, as they currently resemble internal variable names rather than clear explanations. To enhance the clarity and understanding for readers, I recommend refining the descriptions of these settings. Considering the importance of Table S7 for comprehending the different settings, including it in the main text would be advantageous. This inclusion would ensure that readers have access to the necessary information when exploring the various settings discussed in the manuscript. Furthermore, it would be helpful to specify which network was utilized: model or model_ptm. Additionally, providing details about the original settings, such as num_recycles=3 and num_ensemble=1, could be valuable. Consider incorporating this information into Table S7.

Response:

Thank you for the great suggestion. We have made significant changes and improvement on **Table S7** and moved it to the main text as **Table 3**. The description of the table has been revised and improved. The new **Table 3** includes more information, such as AlphaFold2 network used for prediction and AlphaFold2 settings including num_recycles, and num_ensemble.

3. Lastly, it is worth considering whether additional diversity could have been achieved by utilizing both model and model_ptm? Exploring this possibility and discussing its potential impact on the results would further enrich the manuscript.

Response:

Thank you for the suggestion. We performed a new experiment to run AlphaFold2 using the monomer_ptm network with the same parameters (num_recycles=8, num_ensemble=8) as the "default" method participating CASP15 to predict the structures of the 62 CASP15 targets. The results show that there is no significant difference between the "monomer_ptm" network and the "monomer" network on average as their per-target average maximum TM-scores are 0.7968 and 0.7981, respectively, which confirms the claim from the AlphaFold2 GitHub repository that monomer_ptm network is slightly less accurate. However, using monomer_ptm network can generate better models for some targets (e.g., for target T1119, 0.9583 TM-score of top1 model generated by "monomer_ptm" compared to 0.8816 TM-score of the top1 model generated by "monomer"). This demonstrates that even though there is no significant difference on average, using both monomer_ptm and monomer can generate more diverse models and can improve model quality for some targets.

Therefore, incorporating monomer_ptm network in the model generation process can enhance the diversity of the models and the quality of top1 model. We have added the discussion of the new results into the **Discussion section**.

4. Figure 2 shows the best models generated by different sampling methods compared to the AlphaFold baseline. What does a similar plot for rank 1 look like? In addition, the combine method is clearly sampling more structures than the AlphaFold baseline; thus, using the best model is not a fair comparison in this case. A larger pool of models will have a higher chance of generating a better model. It should be clear how many models the maximum is selected from, and for a comparison against the baseline it can only be for the rank 1.

Response:

Thank you for the great suggestion. We have added the performance analysis of rank 1 (top 1) model selected by AlphaFold2's pLDDT score (new **Figure 2B**) into the section "**The importance of sampling more models using AlphaFold2 with diverse MSA and template inputs**". The average TM-score of the top1 models generated by the "combine" method for 62 common targets is 0.786, higher than 0.77 of NBIS-AF2-standard without significant difference (p-value = 0.05938) according to one-sided Wilcoxon signed rank test. In **Figure 2C**, the TM-score difference between the top1 model of the "combine" method selected by AlphaFold2's pLDDT score and the top1 model of NBIS-AF2-standard as well as the number of models of the "combine" method is reported for each target. On most targets, the difference is positive, i.e., the top1 model of the "combine" method has a higher TM-score than NBIS-AF2-standard. As shown in **Figure 2B**, four sampling methods (i.e., *default*, *default_seq_temp*, *original* and *ori_seq_temp*) have higher average TM-score of top1 models than NBIS-AF2-standard on the common targets, but there is no significant difference between each of them and NBIS-AF2-standard.

5. However, another more interesting question than improving over the baseline is which settings are best, and that is not even tested. Most settings could be compared head-to-head on a common set of 58 (?) targets instead.

Response:

Thanks for the great suggestion. We have added new **Figure 2D** in section "**The importance of sampling more models using AlphaFold2 with diverse MSA and template inputs**" to plot the TM-scores of the different sampling methods ("*default*", "*default_seq_temp*", "*original*", "*ori_seq_temp*", "*colabfold*", "*colab_seq_temp*") on the common 58 targets ordered by their average TM-score of the top1 model. The results of the one-sided Wilcoxon test indicate that there is no significant difference between the four better performing methods "*original*" and "*ori_seq_temp*", "*original*" and "*default_seq_temp*" in terms of the TM-score of the top1 models. However, there is significant difference between "*original*" and each of the other three sampling methods (*default*, *colabfold*, and *colab_seq_temp*), with the p-value of 0.04604, 0.01678 and 0.01846 respectively.

6. How different are the distributions for the different sampling techniques for each target, i.e, the combination of all settings seems to generate a diverse set of models, but are they only sampling “one” conformation internally?

Response:

Thanks for the great question. We conducted a new analysis in section “**The importance of sampling more models using AlphaFold2 with diverse MSA and template inputs**” to quantify the distribution of models generated by the different methods. For this purpose, we calculated the average pairwise similarity score (PSS) of the models produced by each method. A higher average PSS indicates that the models in the model pool are very similar (e.g., in one conformation), while a lower PSS suggests the likely presence of multiple conformations in the model pool.

To visualize the results, the average PSS of the models from “*combine*”, “*default*”, “*default_seq_temp*”, “*original*”, “*ori_seq_temp*”, “*colabfold*” and “*colab_seq_temp*” are plotted for each of the 58 common targets in **Figure 2E**. In case the average values are almost the same, the dot denoting the average PSS of the “*combine*” model pool is always plotted at the top, which may cover the dots denoting the almost same values of the other methods. The figure shows that, for 33 out of 58 targets, the average PSS of the methods is greater than 0.9, indicating that they all generated models of similar/same conformations. However, for 22 out of 58 targets, the average PSS of the “*combine*” model pool is less than 0.8, suggesting there is a diverse set of models in the model pool. Interestingly, for T1104, T1179, T1119, T1123, T1178 and T1154, some methods generated very similar conformations (e.g., the average PSS of “*colabfold*” models generated for T1104 is 0.8751), while the other methods (e.g., “*default_seq_temp*”) generated models with more different conformations (the average PSS of “*default_seq_temp*” for T1104 is 0.6247). The diversity of the MSAs and templates used by the different methods increases the variety of models in the “*combine*” model pool for these targets. The average TM-score of top1 models of the “*combine*” method on the 33 targets that has PSS value greater than 0.9 is 0.8858, only slightly higher than 0.8823 of NBIS-AF2-standard, indicating that the targets are mostly easy and generating more models from diverse MSAs and templates only have a small effect on the performance. However, on the 22 targets for which the PSS value of top1 models of the “*combine*” method is less than 0.8, the average TM-score of the top1 models of the “*combine*” is 0.6344, notably higher than 0.6073 of NBIS-AF2-standard, indicating that these targets are mostly harder ones and generating more models from diverse MSAs and templates has a larger effect on the performance. Indeed, for 14 out of the 22 targets, the top1 model in the “*combine*” model pool has a higher TM-score than that of NBIS-AF2-standard, showing that the increased variety of models improves the quality of the models for 63.64% of these targets.

7. What about not using any templates? Sometimes a wrong template can confuse the method and produce a wrong model. You should test: no-template, pdb70, PDB_sort90.

Response:

Thank you for the great suggestion. According to your comment, we conducted additional experiments with AlphaFold2 without using any template as input. We used the same parameters

values as the "default" and "default_seq_temp" methods to generate five models without any templates (called "default_no_template") for each of the 58 common targets. The resulting average top1 TM-scores for the three methods are 0.7853 for "default_no_template", 0.783 for "default", and 0.7873 for "default_seq_temp", respectively. A statistical analysis using the one-sided Wilcoxon test shows that there is no significant difference between these average top1 TM-scores. This finding indicates that, without utilizing templates, the performance of the models remains comparable to those obtained with the "default" and "default_seq_temp" sampling methods on average. Furthermore, without using any template information, AlphaFold2 can generate better structure for some targets (e.g., 0.9619 TM-score of the top1 model generated by "default_no_template", higher than 0.8816 TM-score of the top1 model generated by "default" for target T1119). Therefore, incorporating this sampling method without using templates into the model generation process can enhance the diversity of the models. We have added the comparison in the discussion section.

8. The combine method from which settings are the rank1 models originating?

Response:

Thanks for the question. In section "The importance of sampling more models using AlphaFold2 with diverse MSA and template inputs", new **Figure 2C** illustrates the original source of rank1 (top1) model of the "combine" method for each target (each source denoted by a different color), the number of models in the "combine" method for each target, and the TM-score difference between the top1 model selected by AlphaFold2 pLDDT score in the "combine" method and the top1 model of NBIS-AF2-standard. As shown in Figure 2C, rank1 model originated from different sources.

9. The quality assessment should be analyzed using selected model quality for each target, like Figure 2 on rank 1 (selected) models or scatterplots with correlations. Are the reported correlations overall or per-target?

Response:

Thanks for the suggestion. We have replaced **Table 4** with new **Figure 7** in section **Comparison of quality assessment methods of ranking structural models** to provide a clearer illustration of the TM-score of the selected rank1 (top1) model for each target and the correlations between the predicted quality scores and the true quality scores for each target. The description of the results been changed accordingly. The reported correlations or TM-scores of rank1 models are per-target.

10. Using consensus or comparisons of server/human models is very limited outside the CASP setting, so it would be better if the authors could discuss the performance of a QA that only uses internally generated models that can be reproduced outside the setting of CASP, or single-model QAs.

Response:

Thanks for the great comment. We would like to clarify that all the server and human models (*server_model_dataset* and *human_model_data_set*) to evaluate the quality assessment methods including the consensus methods in our manuscript were all generated by our in-house MULTICOM prediction system. *server_model_dataset* and *human_model_data_set* do not contain any models predicted by third-party predictors. Therefore, the results / performance of the quality assessment methods can be reproduced outside the setting of CASP. According to the results on our internally generated models, the consensus method is useful in the real-world prediction setting where dozens of models are generated for a target. Different from previous CASP competitions, CASP15 did not release the server models predicted by the CASP predictors for human predictions during the CASP15 experiment. Therefore, all the results of our methods for tertiary structure prediction and quality assessment were based on the structural models predicted by our internal system. Therefore, the quality assessment methods used in our system can be replicated or reused by users in their internal prediction system. We added a sentence “It is worth noting that the models in both the *server_model_dataset* and *the human_model_dataset* were all internally predicted by our in-house MULTICOM system without including any models predicted by the third-party predictors” into the manuscript to emphasize that the models used to evaluate the quality assessment methods were all generated internally.

11. The authors claim they are using sampling, but the sampling is very minor as each setting only generated five models; thus, with 8 settings, 40 different models are generated. Some of the more successful groups in CASP15 generated thousands of models using dropout to improve the diversity of models. Is that something that could be used here as well?

Response:

Thank you for the great point. Indeed, in the CASP15 experiment, our predictors only sampled dozens of models for each target in contrast thousands of models generated by a few other more successful predictors. Yet, the performance of our predictors is very close to the best tertiary structure predictors that had a lot of GPUs to generate many more models, indicating that our sampling method based on diverse MSAs and templates is very efficient and effective. Therefore, our method is very suitable for users who have limited GPU resource and can only generate a small number of models for each target. However, sampling thousands of models using dropout to improve the diversity of models is useful, as demonstrated by some other top predictors in the CASP15 experiment. Combining our sample method and the dropout-based sampling tested by some external CASP predictors such as the Wallner human predictor to generate thousands of models should further improve the prediction performance. We add the discussion about this issue into the Discussion section.

12. A related question is whether the MSAs could be resampled, assuming they are large, instead of generating new ones against different databases. Could too many sequences be a problem?

Response:

Thanks for the insightful question. When dealing with large MSAs, resampling is indeed possible and can be useful. In fact, AlphaFold2 internally employs a resampling process on input MSAs to generate diverse MSAs for inference. The sequences in the input MSAs will go through a “MSA clustering” process to generate 512 MSA clusters, which are then used as input feature “msa_feat” for AlphaFold2 models to make predictions. For the sequences not selected as MSA clusters, they are not fully excluded but randomly sampled, typically with 1024 or 5120 samples, and are treated as an additional input feature called “extra_msa_feat” for AlphaFold2. AlphaFold2 will sample $\text{num_recycles} * \text{num_ensemble}$ times of “msa_feat” and “extra_msa_feat” during the inference time, which significantly enhances the robustness and the diversity of the prediction. In summary, MSAs can be resampled, and the resampling is used by AlphaFold2 internally. This internal resampling of MSAs by AlphaFold2 is complementary with our external MSA/template sampling because our approach may find some sequences that AlphaFold2’s MSA generation may not find.

13. Concerning the Foldseek structure alignment-based refinement. How did it actually perform? If the mean performance did not change, and some targets were significantly improved. It implies that some targets also got significantly worse (otherwise, the mean would also be higher). Either report the performance or remove it; since it was only used for 27 targets and the performance gain is weak, it does not need to be highlighted in the abstract.

Response:

Thanks for the good suggestion. According to your suggestion, we have added **Table S7** into the supplementary document to report the detailed performance (the average GDT-TS of the original models and that of the refined models) of the refinement method on each target. On 10 of 23 targets tested, the refined models have higher or equal average GDT-TS.

14. For the T1180 case in Figure 3, are the four templates found by Foldseek more or less identical? And how similar are they to the one found by the sequence search? It would be clearer if the rainbow coloring is based on the target sequence, but it seems as if the templates only cover the N-terminal part.

Response:

Thank you for the great question. The four templates found by Foldseek are 3JS3A, 3JS3B, 3NNTB, 4H3DB. The last template (4H3DB) can be found by the sequence-based template searching. The similarities between the four templates are larger than 0.95 TM-score calculated by TMAAlign. So, they are quite similar. Yes, the templates only cover one domain of the two-domain target.

15. Would it have worked by just injecting the sequence-found template four times, effectively increasing the weight on the templates? Again, the paragraph has a “may be the reason...”; please check the reason. In addition, by comparing the new templates in Figure 3, it seems as if the model improvement is in the part that is not in the templates, or am I wrong? If that is the

case, more sampling/recycles is more likely the reason for the improvement rather than a better template found by Foldseek. Please, comment on this.

Response:

Thanks for the insightful questions and suggestions. To rigorously verify the claims and find the true cause of the improvement of the refinement on T1180, as you suggested, we performed the following three post-CASP15 experiments with AlphaFold2 to generate 15 models respectively, which was the total number of models produced / used during the refinement process. The refinement process used the five models of the “*default*” sampling method as initial models to generate 5 models in each of the three refinement iterations, resulting in 15 refined models in total for selection.

In the first experiment, the “*default*” sampling method was used to generate 15 models for T1180. The GDT-TS of the top1 model is 0.8583, higher than 0.7322 of the top1 model among the five models initially generated by the “*default*” sampling method, but lower than 0.8951 of the final refined model from the refinement. The experiment indicates that increased sampling can yield models of higher quality, but still cannot reach the quality of the refinement process.

In the second experiment, the combined alignments in each iteration of the refinement process along with the sequence search-found templates were fed to AlphaFold2 to generate 15 models. The GDT-TS of the top1 models for each iteration is 0.666, 0.8661 and 0.8806. The final score of 0.8806 is very close to 0.8951 of the final refined model submitted to CASP15, demonstrating that iteratively adding Foldseek-found structure alignments into the MSA in the refinement process is a reason for the improvement in model quality.

In the third experiment, the four templates (3JS3A, 3JS3B, 3NNTB, 4H3DB) identified by Foldseek along with the initial MSA were provided to AlphaFold2 to generate 15 models. The GDT-TS of the top1 model of the refinement process is 0.71, much lower than 0.8951 of the final refined model submitted to CASP15, indicating that the structural templates is not the reason leading to the model quality improvement. Furthermore, we used one template 4H3DB four times as structural templates for AlphaFold2 to generate models, resulting in a top1 model with GDT-TS of 0.7231, much lower than 0.8951 of the final refined model submitted to CASP15. This further confirms that adding more templates into the AlphaFold2 model generation is not the reason that the refinement process produced the high-quality models in CASP15.

The three experiments above shown that the causes of the better quality of refined models are the addition of Fold-seek based alignments into the MSA and the sampling of more models. Adding more templates does not contribute to the improvement on this target. The new analysis and the correction on the incorrect claim have been added to section “**The effect of Foldseek structure alignment-based refinement method**”.

Minor:

Starting paragraphs with boldface Figure #no is confusing, at least in manuscript form; they look almost like the caption.

Response:

Thank you for pointing out the confusion. We have decreased the font size of the figures and tables in the main text to distinguish the main text from the caption.

Figure 2 caption says GDT, figure says TM.

Response:

We have fixed the problem.

REVIEWERS' COMMENTS:

Reviewer #2 (Remarks to the Author):

All my comments and questions have been addressed.